# Multiaccuracy and Multicalibration via Proxy Groups

**Beepul Bharti** [1 2]   **Mary Versa Clemens-Sewall** [3]   **Paul Yi** [4]   **Jeremias Sulam** [1 2 5]

## Abstract

As the use of predictive machine learning algorithms increases in high-stakes decision-making, it is imperative that these algorithms are fair across sensitive groups. However, measuring and enforcing fairness in real-world applications can be challenging due to missing or incomplete sensitive group information. Proxy-sensitive attributes have been proposed as a practical and effective solution in these settings, but only for parity-based fairness notions. Knowing how to evaluate and control for fairness with missing sensitive group data for newer, different, and more flexible frameworks, such as multiaccuracy and multicalibration, remains unexplored. In this work, we address this gap by demonstrating that in the absence of sensitive group data, proxy-sensitive attributes can provably be used to derive actionable upper bounds on the true multiaccuracy and multicalibration violations, providing insights into a predictive model's potential worst-case fairness violations. Additionally, we show that adjusting models to satisfy multiaccuracy and multicalibration across proxy-sensitive attributes can significantly mitigate these violations for the true, but unknown, sensitive groups. Through several experiments on real-world datasets, we illustrate that approximate multiaccuracy and multicalibration can be achieved even when sensitive group data is incomplete or unavailable.

[1]Department of Biomedical Engineering, Johns Hopkins University, Baltimore, USA [2]Mathematical Institute of Data Science, Johns Hopkins University, Baltimore, USA [3]Department of Applied Mathematics & Statistics, Johns Hopkins University, Baltimore, USA [4]St. Jude Children's Research Hospital, Arlington, USA [5]Department of Computer Science, Johns Hopkins University, Baltimore, USA. Correspondence to: Beepul Bharti <bbharti1@jhu.edu>, Jeremias Sulam <jsulam1@jhu.edu>.

*Proceedings of the $42^{nd}$ International Conference on Machine Learning*, Vancouver, Canada. PMLR 267, 2025. Copyright 2025 by the author(s).

## 1. Introduction

Predictive machine learning algorithms are increasingly being used in high-stakes decision-making contexts such as healthcare (Shailaja et al., 2018), employment (Freire & de Castro, 2021), credit scoring (Thomas et al., 2017), and criminal justice (Rudin et al., 2020). Although these predictive models demonstrate impressive overall performance, growing evidence indicates that they can often exhibit biases and discriminate against certain sensitive groups (Obermeyer & Mullainathan, 2019; Dastin, 2022; Li et al., 2023). For instance, ProPublica's investigation (Angwin et al., 2022) revealed significant racial disparities in recidivism risk assessment algorithms, which disproportionately classified African Americans as high-risk for re-offending. As the deployment of these algorithms increases, regulatory bodies worldwide, including the US Office of Science and Technology Policy (OSTP) (of Science & Policy, 2022), European Union (Commission, 2021), and United Nations (United Nations Educational & , UNESCO), have emphasized the importance of ensuring that predictive algorithms avoid discrimination and uphold fairness.

These concerns have led to the emergence of *algorithmic fairness*, a field dedicated to ensuring that predictive models do not inadvertently discriminate against sensitive groups defined by sensitive attributes such as race, age, or biological sex. Unfortunately, measuring and controlling a model's fairness can be challenging in many real-world settings, as sensitive group information is often incomplete or unavailable (Holstein et al., 2019; Garin et al.; Yi et al., 2025). In certain contexts, like healthcare, privacy and legal regulations such as the HIPAA Privacy Rule restrict access to sensitive data. In other cases, the information was not collected because it was considered unnecessary (Weissman & Hasnain-Wynia, 2011; Fremont et al., 2016; Zhang, 2018). Despite these obstacles, it remains crucial to evaluate a model's fairness before and during its deployment. This raises the question: how can we evaluate and promote fairness when sensitive group information is imperfect or missing altogether?

One popular approach, widely applied in healthcare (Brown et al., 2016), finance (Zhang, 2018), and politics (Imai & Khanna, 2016) is to utilize proxy attributes in place of true attributes. Proxy methods have been immensely effective

in evaluating and controlling for traditional *parity*-based notions of fairness (Diana et al., 2022; Bharti et al., 2024; Awasthi et al., 2021; Prost et al., 2021; Zhao et al., 2022; Awasthi et al., 2020; Gupta et al., 2018; Kallus et al., 2022), such as demographic parity (Calders et al., 2009), equalized odds (Hardt et al., 2016), and disparate mistreatment (Zafar et al., 2017), which all aim to equalize model statistics across protected groups.

While enforcing parity is desirable in some settings, it can also lead to undesirable trade-offs. For instance, in breast cancer screening, incidence rates vary by age, with older women generally at higher risk than younger women (Kim et al., 2025). Thus, equalizing a model's false positive rates across age groups might reduce the sensitivity of cancer detection in older women, who are more likely to have the disease. Conversely, equalizing false negative rates might lead to unnecessary biopsies by increasing false positive rates in certain groups. Instead, a more appropriate fairness criterion would be to ask that the model's risk predictions approximately reflect true probabilities within each age group.

These types of domain-specific challenges have led to the development of two newer fairness notions: multiaccuracy and multicalibration (Kim et al., 2019; Gopalan et al., 2022; Hébert-Johnson et al., 2018). Instead of enforcing parity, these methods ensure that model predictions are unbiased and well-calibrated across groups. They can be applied to complex, overlapping sensitive groups—such as those defined by race and gender—while maintaining high predictive accuracy and ensuring the model remains useful in practice, offering significant advantages over traditional parity-based metrics. As a result, enforcing multiaccuracy or multicalibration is powerful and often preferable in many contexts (Kim et al., 2019; Hébert-Johnson et al., 2018). However, a key challenge remains: when sensitive group data is missing, how can we build provably multiaccurate and multicalibrated models leveraging proxies?

Tackling this issue is essential for developing models that are fair across multiple complex groups without sacrificing accuracy or utility. In this work, we address this gap. We study how to estimate multiaccuracy and multicalibration fairness violations without access to true sensitive attributes. We show that proxy-sensitive attributes can be used to derive computable upper bounds on these violations, capturing the model's *worst-case* fairness. Additionally, we demonstrate that post-processing a model to satisfy multiaccuracy or multicalibration across proxies effectively reduces the worst-case fairness violations, offering practical insights. In conclusion, we demonstrate that even when sensitive information is incomplete or inaccessible, proxies can greatly help in providing *approximate* multiaccuracy and multicalibration protections in a useful and meaningful way.

## 1.1. Related Work

Using proxy-sensitive attributes to measure and enforce model fairness has been extensively studied for various parity-based fairness notions.

**Measuring fairness.** Measuring a model's true fairness through proxies has become an important area of research. Chen et al. (2019) were among the first to tackle this challenge by studying the error in measuring demographic parity using proxies derived from thresholding the Bayes optimal predictor for the sensitive attribute. Awasthi et al. (2021) focus on equalized odds, identifying key properties that proxies must satisfy to accurately estimate true equalized odds disparities. Kallus et al. (2022) further examine the ability to identify traditional parity-based fairness violations. They demonstrate that, under general assumptions about the distribution and classifiers, it is usually impossible to pinpoint fairness violations accurately using proxies. Additionally, by assuming access to the Bayes optimal predictor for the sensitive attribute, they provide tight upper and lower bounds on various fairness criteria, thereby characterizing the feasible regions for these violations. More generally, considering any proxy model instead of the Bayes optimal, Zhu et al. (2023) shows that estimating true parity-based fairness disparities using proxies results in errors proportional to the proxy error and the true fairness disparity. Most recently, Bharti et al. (2024) address a setting with more limited information compared to Zhu et al. (2023), providing computable and actionable upper bounds on true equalized odds disparities based on the proxy's misclassification error and proxy group-wise predictor statistics.

**Enforcing fairness.** An equally important question is how to ensure fairness using proxies. Awasthi et al. (2020) examine the post-processing method for equalized odds (Hardt et al., 2016) when noisy proxies are used instead of true sensitive attributes. They show that, under conditional independence assumptions, using proxies in the post-processing method results in a predictor with reduced equalized odds disparity. Wang et al. (2020), working with a slightly different noise model, propose robust optimization approaches to train fair models using noisy sensitive features. Having proven that fairness violations are often unidentifiable, Kallus et al. (2022) take a different approach and focus on reducing the worst-case violations. Under additional smoothness assumptions they derive tighter feasible regions for fairness disparities, offering improved worst-case guarantees for fairness violations. More recently, Bharti et al. (2024) characterize the predictor that has optimal worst-case violations and provide a generalized version of Hardt et al. (2016)'s method that returns such a predictor. Taking a different perspective, Lahoti et al. (2020) avoid relying on proxies altogether. Instead, they propose solving a minimax optimization problem over a vast set of subgroups, reasoning

that any good proxy for a sensitive feature would naturally be included in this set. Diana et al. (2022) address the problem of learning proxies that enable downstream model developers to train models that satisfy common parity-based fairness notions. They demonstrate that this entails constructing a multiaccurate proxy and introduce a general oracle-efficient algorithm to learn such proxies.

## 1.2. Our Contributions

There exists a rich line of work that studies how to evaluate and enforce parity-based notions of fairness when sensitive attribute data is missing via proxies. These, however, do not extend to settings where multiaccuracy and multicalibration are more appropriate, limiting their applicability in data-scare regimes. In this work, we address this issue. Our main contributions are the following:

1. We study the problem of estimating multiaccuracy and multicalibration violations of a predictive model without access to sensitive group information.

2. We derive computable upper bounds for multiaccuracy and multicalibration violations using proxy-sensitive attributes.

3. We show that post-processing a model to satisfy multiaccuracy and multicalibration across proxies reduces the worst-case violations, allowing us to provide meaningful fairness guarantees without access to sensitive group data.

**Organization.** The remainder of the paper is structured as follows. In Section 2, we introduce the necessary notation and formalize the setting. Section 3 provides background on multiaccuracy and multicalibration. Our main theoretical results are presented in Section 4 and Section 5, where we establish computable upper bounds on the multiaccuracy and multicalibration violations and demonstrate how to minimize them. Experimental results are detailed in Section 6. Finally, in Section 7 we discus the implications of our work and provide closing remarks.

## 2. Preliminaires

**Notation.** We consider a binary classification setting [1] with a data distribution $\mathcal{D}$ supported on $\mathcal{X} \times \mathcal{Z} \times \mathcal{Y}$, where $\mathcal{X} \subseteq \mathbb{R}^d$ and $\mathcal{Z} \subseteq \mathbb{R}^K$ represent a $d$-dimensional feature space and $K$-dimensional sensitive group space, respectively, and $\mathcal{Y} = \{0, 1\}$ denotes the binary label space. For an individual represented by the pair $(X, Z)$, $X$ is a vector of features and $Z$ is a vector of sensitive features (e.g., race, biological, age). We denote $\mathcal{G} = \{g : \mathcal{X} \times \mathcal{Z} \mapsto \{0, 1\}\}$ as the set

---

[1]One can also extend this to a $K$-class problem using a one-vs-all approach.

---

of functions that define complex, potentially intersecting groups in $\mathcal{X} \times \mathcal{Z}$. For any $g \in \mathcal{G}$, $g(X, Z) = 1$ indicates that the individual $(X, Z)$ belongs to group $g$. For example, let $X_1$ be an individual's credit score and let $Z_1$ and $Z_2$ represent the individual's age and membership in the African American group. Then, $g(X, Z) = \mathbf{1}\{X_1 > 700 \wedge Z_1 > 40 \wedge Z_2 = 1\}$ specifies the group of all African Americans over the age of 40 with a credit score over 700. In this way, it is easy to define arbitrary overlapping groups defined by the intersection of basic attributes and other features. Finally, in our setting, a model is a function $f : \mathcal{X} \to R$ that maps from the feature space to some discrete domain $R \subseteq [0, 1]$. We denote its image as $\text{Im}(f) = \{f(X) : X \in \mathcal{X}\}$ and assume that $|\text{Im}(f)| < \infty$.

**Problem Setting.** In this work, the primary objective is to assess whether a model $f$ is fair with respect to a set of sensitive groups $\mathcal{G}$ *without* having access to the functions in $\mathcal{G}$ to determine group membership. Formally, and similar to previous work (Awasthi et al., 2021; Kallus et al., 2022), we consider a setting where we do not have access to samples $(X, Z, Y)$ from the complete distribution $\mathcal{D}$ – and thus we are unable to use the true set of grouping functions $\mathcal{G}$ since their domain is supported $\mathcal{X} \times \mathcal{Z}$. Instead, we assume access to a sufficient number of samples $(X, Y)$ from $\mathcal{D}_{\mathcal{X}\mathcal{Y}}$, the marginal distribution over $\mathcal{X} \times \mathcal{Y}$, which allow us to reliably evaluate the overall performance of the predictor $f$ via its mean-squared error

$$\text{MSE}(f) = \mathop{\mathbb{E}}_{(X,Y) \sim \mathcal{D}_{\mathcal{X}\mathcal{Y}}} [(Y - f(X))^2]. \quad (1)$$

Additionally, we assume a proxy developer that has access to samples $(X, Z)$ from $\mathcal{D}_{\mathcal{X}\mathcal{Z}}$, the marginal distribution over $\mathcal{X} \times \mathcal{Z}$. They provide a set of learned proxy functions $\hat{\mathcal{G}} = \{\hat{g} : \mathcal{X} \mapsto \{0, 1\}\}$ for $\mathcal{G}$ that only use features $X$, allowing us to determine *proxy* group membership. Moreover, via the proxy developer, we know how well any proxy $\hat{g} \in \hat{\mathcal{G}}$ approximates its associated true $g \in \mathcal{G}$ through its misclassification error,

$$\text{err}(\hat{g}) = \mathop{\mathbb{P}}_{(X,Z) \sim \mathcal{D}_{\mathcal{X}\mathcal{Z}}} [\hat{g}(X) \neq g(X, Z)]. \quad (2)$$

Through this setup, we are modeling real-world situations where we lack information about individuals' basic sensitive attributes, such as sex and race, preventing us from accurately identifying individuals' membership in complex intersecting groups (for example, white women over the age of 40). Instead, we rely on proxies to (approximately) represent all groups. Importantly, we do not make stringent, unverifiable assumptions about these proxy functions, unlike other studies (Prost et al., 2021; Awasthi et al., 2020; 2021); we only consider knowing their error rates, $\text{err}(\hat{g})$.

With our setting fully described, we now turn to our main objective: assessing the fairness of the model $f$ with re-

spect to $\mathcal{G}$. In this work we focus on two fairness concepts—*multiaccuracy* and *multicalibration*—which we now formally define.

## 3. Multiaccuracy and Multicalibration

**Multiaccuracy**. Multiaccuracy (MA) is a notion of fairness originally introduced by (Kim et al., 2019; Hébert-Johnson et al., 2018). For any sensitive group $g \in \mathcal{G}$, MA evaluates the bias of a model $f$, conditional on membership in $g$ via

$$\mathsf{AE}_{\mathcal{D}}(f, g) = \left| \mathbb{E}_{\mathcal{D}}[g(X, Z)(f(X) - Y)] \right| \qquad (3)$$

and requires that $\mathsf{AE}_{\mathcal{D}}(f, g)$ be small for all groups $g \in \mathcal{G}$.

**Definition 3.1** (Multiaccuracy (Kim et al., 2019))**.** Fix a distribution $\mathcal{D}$ and let $\mathcal{G}$ be a set of groups. A model $f$ is $(\mathcal{G}, \alpha)$-multiaccurate if

$$\mathsf{AE}_{\mathcal{D}}^{\max}(f, \mathcal{G}) := \max_{g \in \mathcal{G}} \mathsf{AE}_{\mathcal{D}}(f, g) \leq \alpha \qquad (4)$$

$(\mathcal{G}, \alpha)$-MA requires that the predictions of $f$ be approximately unbiased overall *and* on every group. Building on this, (Hébert-Johnson et al., 2018) introduced a stronger notion of group fairness known as multicalibration (MC), which demands unbiased *and* calibrated predictions. Central to evaluating MC is the expected calibration error (ECE) for a group $g \in \mathcal{G}$

$$\mathsf{ECE}_{\mathcal{D}}(f, g) = \mathbb{E}_{v \sim \mathcal{D}_f} \left| \mathbb{E}[g(X, Z)(f(X) - Y)|f(X) = v] \right|$$

where $\mathcal{D}_f$ is the distribution of the predictions made by $f$ under $\mathcal{D}$. MC requires that $\mathsf{ECE}_{\mathcal{D}}(f, g)$ be small for all groups $g \in \mathcal{G}$.

**Definition 3.2** (Multicalibration (Hébert-Johnson et al., 2018))**.** Fix a distribution $\mathcal{D}$ and let $\mathcal{G}$ be a set of groups. A model $f$ is $(\mathcal{G}, \alpha)$-multicalibrated if

$$\mathsf{ECE}_{\mathcal{D}}^{\max}(f, \mathcal{G}) := \max_{g \in \mathcal{G}} \mathsf{ECE}_{\mathcal{D}}(f, g) \leq \alpha. \qquad (5)$$

This is an $l_1$ notion of MC as studied in (Gopalan et al., 2022). There also exist $l_2$ (Globus-Harris et al., 2023b) and $l_\infty$ (Hébert-Johnson et al., 2018) variants. $(\mathcal{G}, \alpha)$-MC requires that $f$'s predictions be approximately calibrated on all groups defined by $\mathcal{G}$. Note, MC is stronger than MA because intuitively MC requires MA on every level set of $f$.

Having presented these definitions, the problem we face is now clear: Ideally, we would like to evaluate $\mathsf{AE}_{\mathcal{D}}^{\max}$ and $\mathsf{ECE}_{\mathcal{D}}^{\max}$. However, this requires access to samples $(X, Z, Y) \sim \mathcal{D}$ and the functions $\mathcal{G}$, neither of which we assume to have. As alluded to before, no method exists that

can guarantee–let alone, correct–that a predictor is $(\mathcal{G}, \alpha)$-MA/MC in the absence of ground truth groups $\mathcal{G}$. Fortunately, in the following sections, we demonstrate that with proxies, one can successfully circumvent these limitations and still provide meaningful guarantees.

## 4. Bounds on Multigroup Fairness Violations

We will now demonstrate that is still possible to derive computable and useful upper bounds for the MA and MC violations of a model $f$ across the true groups $\mathcal{G}$, even in the absence of true group information.

Our first result provides computable upper bounds on $\mathsf{AE}_{\mathcal{D}}(f, g)$ and $\mathsf{ECE}_{\mathcal{D}}(f, g)$ for any group $g$.

**Lemma 4.1.** *Fix a distribution $\mathcal{D}$ and model $f$. For any group $g$ and its corresponding proxy $\hat{g}$,*

$$\mathsf{AE}_{\mathcal{D}}(f, g) \leq \mathsf{F}(f, \hat{g}) + \mathsf{AE}_{\mathcal{D}}(f, \hat{g}) \qquad (6)$$
$$\mathsf{ECE}_{\mathcal{D}}(f, g) \leq \mathsf{F}(f, \hat{g}) + \mathsf{ECE}_{\mathcal{D}}(f, \hat{g}) \qquad (7)$$

*where*

$$\mathsf{F}(f, \hat{g}) = \min \left( \mathsf{err}(\hat{g}), \sqrt{\mathsf{MSE}(f) \cdot \mathsf{err}(\hat{g})} \right). \qquad (8)$$

*Furthermore the upper bounds are tight.*

A proof of this result is provided in Appendix B.1. We now make a few remarks. First, the upper bounds on the true MA/MC violations are tight: there exist non-trivial joint distributions over $(f(X), g(X), \hat{g}(X), Y)$—specifically, those for which $\mathsf{err}(\hat{g}) > 0$ and $\mathsf{MSE}(f) > 0$—such that the bounds are attained with equality. Please see Appendix B.1 for a comprehensive discussion regarding these statements. Second, in our setting, both bounds can be directly evaluated because (1) we know $\mathsf{err}(\hat{g})$ for all $g \in \mathcal{G}$ via the proxy developer, who has access to $\mathcal{G}$ and a sufficient number of samples $(X, Z)$ from $\mathcal{D}_{\mathcal{X}\mathcal{Z}}$ to compute $\mathsf{err}(\hat{g})$; and (2) we can reliably compute $\mathsf{MSE}(f)$, $\mathsf{AE}_{\mathcal{D}}(f, \hat{g})$, and $\mathsf{ECE}_{\mathcal{D}}(f, \hat{g})$ because we have access to a sufficient number of samples $(X, Y)$ from $\mathcal{D}_{\mathcal{X}\mathcal{Y}}$.

This result aligns with intuition, showing how the relationship between a proxy $\hat{g}$ and the model $f$ constrains the maximum possible values of $\mathsf{AE}_{\mathcal{D}}(f, g)$ and $\mathsf{ECE}_{\mathcal{D}}(f, g)$. If the proxy $\hat{g}$ is highly accurate in that it predicts the true group $g$ better than $f$ predicts the true label $Y$, i.e. if

$$\mathsf{err}(\hat{g}) < \mathsf{MSE}(f), \qquad (9)$$

then $\mathsf{F}(f, \hat{g}) = \mathsf{err}(\hat{g})$, and the true violations $\mathsf{AE}_{\mathcal{D}}(f, \hat{g})$ and $\mathsf{ECE}_{\mathcal{D}}(f, \hat{g})$ are approximately bounded by their proxy estimates. Conversely, if $f$ is highly accurate in predicting the label $Y$ but the proxy is weaker, so that

$$\mathsf{MSE}(f) < \mathsf{err}(\hat{g}), \qquad (10)$$

---

**Algorithm 1** Multiaccuracy Regression

1: **Input:** Initial model $f$ and set of groups $\mathcal{G}$
2: Solve

$$\hat{f} \in \underset{\lambda \in \mathbb{R}^{|\mathcal{G}|}}{\arg\min}\ \mathsf{MSE}(\hat{f})$$

$$\text{s.t. } \hat{f}(X, Z) = f(X) + \sum_{g \in \mathcal{G}} \lambda_g \cdot g(X, Z)$$

3: Return $\hat{f}$

---

then $\mathsf{F}(f, \hat{g}) = \sqrt{\mathsf{MSE}(f) \cdot \mathsf{err}(\hat{g})}$, and one can attain a better bound by considering a factor of $\sqrt{\mathsf{MSE}(f) \cdot \mathsf{err}(\hat{g})}$ instead of $\mathsf{err}(\hat{g})$. Naturally, as $\mathsf{MSE}(f)$ decreases, the maximum possible values of $\mathsf{AE}_{\mathcal{D}}(f, \hat{g})$ and $\mathsf{ECE}_{\mathcal{D}}(f, \hat{g})$ decreases as well.

Most importantly, with this result, we can provide an upper bound for the true MA and MC violations of $f$ across $\mathcal{G}$.

**Theorem 4.2.** *Fix a distribution $\mathcal{D}$ and model $f$. Let $\mathcal{G}$ be a set of true groups and $\hat{\mathcal{G}}$ be its associated set of proxies. Then, $f$ is $(\mathcal{G}, \beta(f, \hat{\mathcal{G}}))$-MA and $(\mathcal{G}, \gamma(f, \hat{\mathcal{G}}))$-MC where*

$$\beta(f, \hat{\mathcal{G}}) = \max_{\hat{g} \in \hat{\mathcal{G}}} \mathsf{F}(f, \hat{g}) + \mathsf{AE}_{\mathcal{D}}(f, \hat{g}) \tag{11}$$

$$\gamma(f, \hat{\mathcal{G}}) = \max_{\hat{g} \in \mathcal{G}} \mathsf{F}(f, \hat{g}) + \mathsf{ECE}_{\mathcal{D}}(f, \hat{g}) \tag{12}$$

This result directly follows from Lemma 4.1, with a complete proof provided in Appendix B.2. It is particularly valuable because, even without directly evaluating the quantities of interest, $\mathsf{AE}_{\mathcal{D}}^{\mathsf{max}}(f, \mathcal{G})$ and $\mathsf{ECE}_{\mathcal{D}}^{\mathsf{max}}(f, \mathcal{G})$, we can still evaluate these *worst-case* violations, which offers practical utility. For instance, if we need to ensure that $f$ is $(\mathcal{G}, \alpha)$-MC before deployment, we can proceed confidently even without direct access to $\mathcal{G}$, provided that $\gamma(f, \hat{\mathcal{G}}) < \alpha$. Conversely, if the worst-case violations are large, this suggests that $f$ may *potentially* be significantly biased or uncalibrated for certain groups $g$.

In a scenario where the worst-case violations are large, as model developers, we should pause deployment and ask: if it is the case that $\beta(f, \hat{\mathcal{G}})$ or $\gamma(f, \hat{\mathcal{G}})$ are large, can we reduce them such that we can provide better guarantees on the worst-case MA and MC violations of $f$? We show that it is possible in the following section.

## 5. Reducing Worst-Case Violations

The results from the previous section allow us to upper bound the MA and MC violations using proxies. Now, we show that these violations can be provably reduced, yielding stronger worst-case guarantees. Recall that we have a fixed set of proxies $\hat{\mathcal{G}}$, a model $f$, and access to samples $(X, Y)$.

---

**Algorithm 2** Multicalibration Boosting

1: **Input:** Initial model $f$, set of groups $\mathcal{G}$, and $\alpha > 0$
2: Let $m = \lceil \frac{1}{\alpha} \rceil, t = 0, f_0 := f$
3: **while**

$$\max_{g \in \mathcal{G}} \mathbb{P}[g(X, Z) = 1] \cdot \mathbb{E}[\Delta_{v,g}^2 | g(X, Z) = 1] > \alpha$$

4: Set

$$p_{g,v} = \mathbb{P}[g(X, Z) = 1, f_t(X) = v]$$

$$(v_t, g_t) = \underset{v \in [\frac{1}{m}], g \in \mathcal{G}}{\arg\max}\ p_{g,v} \cdot \Delta_{v,g}^2 \geq \alpha$$

$$S_{v_t, g_t} = \{X \in \mathcal{X} : f_t(X) = v, g_t(X, Z) = 1\}$$

$$\tilde{v}_t = \mathbb{E}[Y | f_t(X) = v, g_t(X, Z) = 1]$$

$$v_t' = \underset{v \in [\frac{1}{m}]}{\arg\min}\ |\tilde{v}_t - v|$$

$$f_{t+1}(X) = \begin{cases} v_t', & \text{if } X \in S_{v_t, g_t} \\ f_t(X) & \text{otherwise} \end{cases}$$

$$t = t + 1$$

5: **end while**
6: Let $T := t, \hat{f} := f_T$
7: Return $\hat{f}$

---

Within the bounds $\beta(f, \hat{\mathcal{G}})$ and $\gamma(f, \hat{\mathcal{G}})$, there are only two quantities we can modify by adjusting $f$: the mean squared error $\mathsf{MSE}(f)$ and either $\mathsf{AE}_{\mathcal{D}}(f, \hat{g})$ or $\mathsf{ECE}_{\mathcal{D}}(f, \hat{g})$. Since we lack access to the true grouping functions $\mathcal{G}$ and samples $(X, Z) \sim \mathcal{D}_{\mathcal{X}\mathcal{Z}}$, we cannot reduce the proxy errors $\mathsf{err}(\hat{g})$. Thus, the question is, what modifications can be made to $f$ such that the updated $\hat{f}$ has smaller bounds? We answer this question for the bound on MC in the following theorem.

**Theorem 5.1.** *Fix a distribution $\mathcal{D}$, initial model $f$, and set of proxy groups $\hat{\mathcal{G}}$. If a model $\hat{f}$ satisfies*

$$\mathsf{ECE}_{\mathcal{D}}^{\mathsf{max}}(\hat{f}, \hat{\mathcal{G}}) < \min_{\hat{g} \in \hat{\mathcal{G}}} \mathsf{ECE}_{\mathcal{D}}(f, \hat{g}) \tag{13}$$

$$\mathsf{MSE}(\hat{f}) \leq \mathsf{MSE}(f) \tag{14}$$

*then, it will have a smaller worst-case MC violation, i.e.*

$$\gamma(\hat{f}, \hat{\mathcal{G}}) \leq \gamma(f, \hat{\mathcal{G}}). \tag{15}$$

An identical result for MA is given in Appendix A and proofs for both results are provided in Appendix B.3. This results simply states that if we can obtain a new model $\hat{f}$ that 1) is $(\hat{\mathcal{G}}, \alpha)$-MC at level $\alpha = \min_{\hat{g} \in \hat{\mathcal{G}}} \mathsf{ECE}_{\mathcal{D}}(f, \hat{g})$ and 2) has smaller $\mathsf{MSE}$, then it is guaranteed to have a smaller worst-case violation. Fortunately, both objectives can be nearly achieved using Algorithm 1, proposed by Gopalan et al. (2022), and Algorithm 2, introduced by Roth (2022), which produce MC and MC predictors, respectively.

| Group | err($\hat{g}$) |
|---|---|
| Black Women | 0.027 |
| White Women | 0.122 |
| Asian | 0.060 |
| Seniors | 0.000 |
| Women | 0.000 |
| Multiracial | 0.047 |

*Table 1.* Proxy errors for ACSIncome

| Group | err($\hat{g}$) |
|---|---|
| Black Women | 0.005 |
| White Women | 0.046 |
| Asian | 0.000 |
| Multiracial | 0.000 |
| Black Adults | 0.000 |
| Women | 0.079 |

*Table 2.* Proxy errors for ACSPubCov

| Group | err($\hat{g}$) |
|---|---|
| Women | 0.027 |
| White | 0.092 |
| Asian | 0.068 |
| Black | 0.039 |
| Asian Men | 0.039 |
| Black Women | 0.020 |

*Table 3.* Proxy errors for CheXpert

Algorithm 1 outlines a simple algorithm for MA. The following theorem establishes that the algorithm produces a model $\hat{f}$ that satisfies MA while also guaranteeing an improvement or no deterioration in MSE.

**Theorem 5.2.** *Fix a distribution $\mathcal{D}$, predictor $f$, and set of groups $\mathcal{G}$. Algorithm 1 returns a model $\hat{f}$ that is $(0, \mathcal{G})$-MA. Moreover,*

$$\mathsf{MSE}(\hat{f}) \leq \mathsf{MSE}(f). \tag{16}$$

A proof of this result can be found in (Detommaso et al., 2024), with finite-sample guarantees discussed in (Roth, 2022). Algorithm 1 updates the model $\hat{f}$ by solving a standard linear regression problem, where the features are the predictions of the initial model $f$ and grouping functions $g$.

While Algorithm 1 solves a optimization problem to generate a MA model in a single step, Algorithm 2 is an iterative method to ensure MC. Algorithm 2 starts by checking if $f$ is $\alpha$-MC via the *group average squared calibration error*

$$\mathbb{E}[\Delta_{v,g}^2 | g(X, Z) = 1] \tag{17}$$

where $\Delta_{v,g} = \mathbb{E}[Y - f(X) | f(X) = v, g(X, Z) = 1]$. If this exceeds $\alpha$, it identifies the conditioning event where the calibration error is the largest and refines $f$'s predictions. It iterates like this until convergence and this process returns a new model $\hat{f}$ that is $\alpha$-MC and has an MSE that is close, potentially even lower, than the MSE of the initial model $f$.

**Theorem 5.3.** *Fix a distribution $\mathcal{D}$, predictor $f$ and set of groups $\mathcal{G}$. Algorithm 2 stops after $T < \frac{4}{\alpha^2}$ rounds and returns model $\hat{f}$ that is $(\sqrt{\alpha}, \mathcal{G})$-MC. Moreover,*

$$\mathsf{MSE}(\hat{f}) \leq \mathsf{MSE}(f) + (1 - T)\frac{\alpha^2}{4} + \alpha \tag{18}$$

A proof, along with with finite-sample guarantees can be found in (Globus-Harris et al., 2023a; Roth, 2022).

Having introduced these algorithms for obtaining multiaccurate and multicalibrated models $\hat{f}$, we now have a direct path to reducing worst-case violations. By applying Algorithm 1 or Algorithm 2 (at an appropriate level $\alpha$) to our initial model $f$ using the proxies $\hat{\mathcal{G}}$—that is, enforcing multiaccuracy or multicalibration with respect to the proxies—we can systematically reduce worst-case violations on the true groups $\mathcal{G}$. This simple yet effective approach ensures allows us to still provide meaningful fairness guarantees and reliable predictions across sensitive subpopulations.

# 6. Experimental Results

We illustrate various aspects of our theoretical results on two tabular datasets, ACSIncome and ACSPublicCoverage (Ding et al., 2021), as well as on the CheXpert medical imaging dataset (Irvin et al., 2019). For the ACS datasets, we use a fixed 10% of the samples as the evaluation set. The remaining 90% of the data is split into training and validation sets, with 60% used for training the model $f$ and proxies $\hat{\mathcal{G}}$ and 30% for adjusting $f$. All reported results are averages over five train/validation splits on the evaluation set. For CheXpert, we use the splits provided by (Glocker et al., 2023) for training, calibration, and evaluation. The results and metrics are computed on the evaluation set. We report results for MC in the main body of the paper and defer those for MA to Appendices C and E because MC is a stronger notion that implies MA. The code necessary to reproduce these experiments is available at https://github.com/Sulam-Group/proxy_ma-mc.

## 6.1. ACS Experiments

For the two following tabular data experiments we use the ACS dataset, a larger version of the UCI Adult dataset. In particular, we use the 2018 California data, which contains approximately 200,000 samples. We follow Hansen et al. (2024) and define multiple sensitive groups $\mathcal{G}$ using basic sensitive attributes $Z$ (e.g., sex and race, which model developers aim not to discriminate towards), along with certain features $X$ (e.g., age). Examples of groups $g \in \mathcal{G}$ include white women and black adults. For both experiments, we simulate missing sensitive attributes by excluding some $Z_i$ from the data we use to train our predictive model $f$. Instead, with an auxiliary dataset of samples $(X, Z)$ and the true set of grouping functions $\mathcal{G}$, we obtain a set of proxy functions $\hat{\mathcal{G}}$ to approximate $\mathcal{G}$.

For our initial model $f$, we report the worst-case violations, which we *can* evaluate in our setting. To demonstrate that

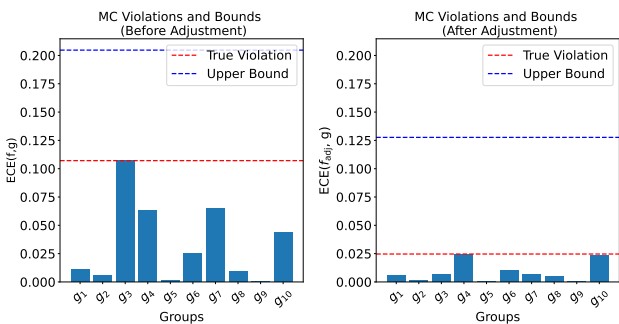

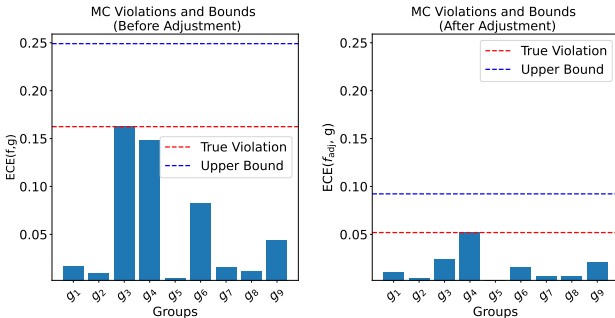

(a) ECE, ECE$^{\text{max}}$ (dotted red line), and worst case violations (dotted blue line) of the original model $f$ and adjusted model $f_{\text{adj}}$ on ACS Income. Here, $f$ is a decision tree.

(b) ECE, ECE$^{\text{max}}$ (dotted red line), and worst case violations (dotted blue line) of the original model $f$ and updated model $f_{\text{adj}}$ on ACSPubcov. Here, $f$ is a decision tree.

enforcing MA and MC with respect to $\hat{\mathcal{G}}$ provably reduces our upper bounds, we apply Algorithm 1 and Algorithm 2 to obtain an adjusted predictor $f_{\text{adj}}$ and report its worst-case violations as well. Additionally, for both the initial model $f$ and adjusted model $f_{\text{adj}}$ we report the AE and ECE, *with respect to the true groups* $g \in \mathcal{G}$, along with their maximums, AE$^{\text{max}}$ and ECE$^{\text{max}}$. Recall that in our setting, we *cannot* actually evaluate these quantities but we report them to showcase that they lie under our bounds, illustrating the validity of our theoretical results. For these tabular experiments, we model $f$ with a logistic regression model, a decision tree, and Random Forest. We report the results for the decision tree in the following sections and defer the remainder to Appendices C and E.

### 6.1.1. ACSINCOME

For this experiment, we consider the task of predicting whether working adults living in California have a yearly income that exceeds \$50,000. Examples of features $X$ include `occupation` and `education`. To simulate missing sensitive attributes, we exclude the `race` attribute.

In Table 1, we report the errors of the learned proxies $\hat{\mathcal{G}}$ for specific groups. For groups that do not depend on `race`, such as `seniors` and `women`, their respective proxies $\hat{g}$ are perfectly accurate, exhibiting zero misclassification error. However, for groups like `multiracial` and `white women`, the proxies exhibit some error, albeit small. This proves to be useful in providing meaningful guarantees on how multiaccurate and multicalibrated the model $f$ may potentially be with respect to the true (but unobserved) groups $\mathcal{G}$.

Figure 1a showcases the utility of our bounds. Notably, the worst-case violation (dotted red line) allows us to certify that the initial model is approximately 0.21-multicalibrated with respect to the true groups $\mathcal{G}$. This is indeed practically useful as it enables practitioners to obtain a certificate on the MC violation without having access to the true sensitive group information. Additionally, the right-hand graph in Figure 1a

highlights the benefit of applying Algorithm 2 and multical-ibrating the initial model $f$ with respect to the proxy groups $\hat{\mathcal{G}}$. After adjusting $f$, the upper bound decreases, allowing us to certify that the resulting model $f_{\text{adj}}$ is approximately 0.13-multicalibrated–a substantial improvement of 38%.

### 6.1.2. ACSPUBLICCOVERAGE (ACSPUBCOV)

In this experiment, we consider the task of predicting whether low-income individuals ($< \$30,000$) , not eligible for Medicare, have coverage from public health insurance. Examples of features $X$ include `age`, `education`, `income`, and more. To simulate missing sensitive attributes, we exclude the `sex` attribute.

In Table 2, we report the errors of the learned proxies $\hat{\mathcal{G}}$ for specific groups. Notably, for groups independent of `sex`, such as `asian` and `multiracial`, the proxies $\hat{g}$ are perfectly accurate, exhibiting zero misclassification error. However, for groups like `black women` and `white women`, the proxies exhibit some error, though they are small. This arises because, although the proxies functions $\hat{g}$ are not explicit functions of `sex` attribute, there exists a feature, `fertility`, that indicates whether an individual has given birth within the past 12 months and serves as a good predictor of `sex`. This is a prime example of a real-world setting where, even though sensitive attributes may be missing, strong proxies can still enable us to determine true sensitive group membership with high accuracy.

In Figure 1b, we show the results of applying Algorithm 2 to multicalibrate the initial model $f$ with respect to the proxies $\hat{\mathcal{G}}$. Notably, our approach allows us to certify that the initial model is approximately 0.25-MC with respect to the true groups $\mathcal{G}$ despite not having access to them. This result high-lights the utility of our method, as it enables practitioners to obtain performance guarantees without needing the true group information. Furthermore, after applying Algorithm 2 to $f$, the resulting model $f_{\text{adj}}$ is certified to be approximately 0.09-MC, thereby providing a stronger guarantee.

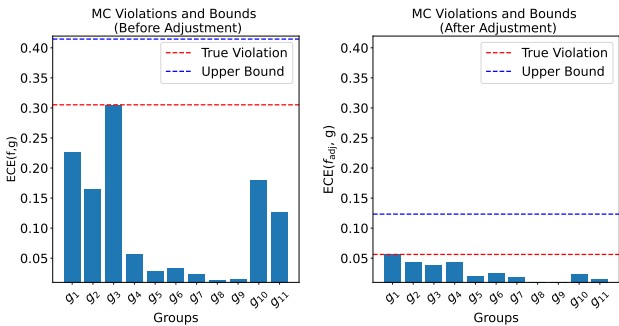

(a) ECE, ECE^max (dotted red line), and worst case violations (dotted blue line) of the original model $f$ and adjusted model $f_{\text{adj}}$ on CheXpert. Here $f$ is a decision tree trained on embeddings of a DenseNet-121 model pretrained on ImageNet.

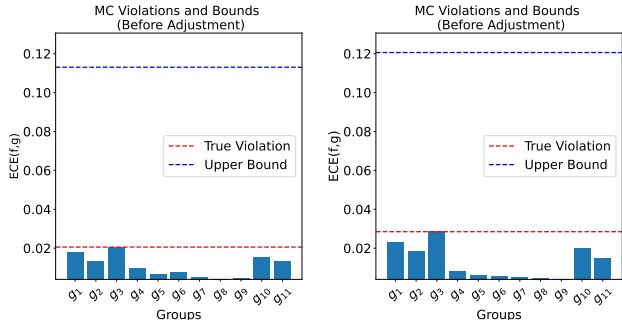

(b) ECE, ECE^max (dotted red line), and worst case violations (dotted blue line) of the original, unadjusted, logistic regression model (left) and end-to-end trained DenseNet-121 model (right).

## 6.2. CheXpert

CheXpert is a large public dataset for chest radiograph interpretation, with labeled annotations for 14 observations (positive, negative, or unlabeled) including `cardiomegaly`, `atelectasis`, and several others. The dataset contains self-reported sensitive attributes including `race`, `sex`, and `age`. Following the set up of Glocker et al. (2023), we work with a sample containing a total of 127,118 chest X-ray scans and consider the task of predicting the presence of `pleural effusion` in the X-rays.

We consider all 14 groups that can be made from conjunctions of `sex` and `race`. Examples of groups $g \in \mathcal{G}$ include `black men`, `asian women`, `white women`, etc. In this example, we assume that we do not have direct knowledge of patient's self-reported `sex` or `race` when training or evaluating our model $f$ (as is common for privacy reasons). Instead, with an auxiliary dataset with samples $(X, Z)$ we use the X-rays to learn proxy functions for `sex` and `race`. We then use them to construct proxies for all conjunctions as well. In Table 3, we report the proxy errors for specific groups.

We consider three different models for $f$. The first is a decision tree classifier trained on features extracted from a DenseNet-121 model (Huang et al., 2017) pretrained on ImageNet (Deng et al., 2009). The second is a linear model (Breiman, 2001) trained on the same features. The third is a DenseNet-121 trained end-to-end on the X-rays.

Figure 2a illustrates the results for the decision tree model. Before any adjustments, our worst-case violation serves as an early warning that the model $f$ may be significantly uncalibrated on certain groups, with a violation as large as $\alpha \approx 0.42$. In a medical setting like this, such a finding is crucial, as it indicates that our predictions could be either overly confident or underconfident on sensitive groups. On the other hand, the right-hand graph of Figure 2a demonstrates the practical benefit of applying Algorithm 2 to multicali-

brate the initial model $f$ with respect to our highly accurate proxies $\hat{\mathcal{G}}$. After a straightforward adjustment, the upper bound on the worst-case violation decreases significantly, certifying that the adjusted model $f_{\text{adj}}$ is approximately 0.13-MC with respect to the true groups.

Figure 2b presents the results for the logistic regression and fully-trained DenseNet models. In these cases, the worst-case violations for both models indicate that they are guaranteed to be approximately 0.11 and 0.12-multicalibrated with respect to the true groups. Notably, both models are approximately 0.03-multicalibrated with respect to the proxies. Thus, further adjustments provide negligible improvements.

## 7. Conclusion

In this work, we address the challenge of measuring multiaccuracy and multicalibration with respect to sensitive groups when sensitive group data is missing or unobserved. By leveraging proxy-sensitive attributes, we derive actionable upper bounds on true the multiaccuracy and multicalibration violations, offering a principled approach to assessing worst-case fairness violations. Furthermore, we demonstrate that adjusting models to be multiaccurate or multicalibrated with respect to proxy-sensitive attributes can significantly reduce these upper bounds, thereby providing useful guarantees on multiaccuracy and multicalibration violations for the true, but unknown, sensitive groups.

Through empirical validation on real-world datasets, we show that multiaccuracy and multicalibration can be approximated even in the absence of complete sensitive group data. These findings highlight the practicality of using proxies to assess and enforce fairness in high-stakes decision-making contexts, where access to demographic information is often restricted. In particular, we illustrate the practical benefit of enforcing multiaccuracy and multicalibration with respect to proxies, providing practitioners with a simple and effective tool to improve fairness in their models.

Naturally, multiaccuracy and multicalibration may not be the most appropriate fairness metrics across all settings. Nonetheless, whenever these notions are relevant, our methods offer, for the first time, the possibility to provide certificates and strengthen them without requiring access to ground truth group data. Lastly, note that without our recommendations of correcting for worst-case fairness with proxies, models trained on this data can inadvertently learn these sensitive attributes indirectly and base decisions on them, leading to potential negative outcomes. Our results and methodology prevent this by providing a principled approach to adjust models and reduce worst-case multiaccuracy and multicalibration violations.

## Impact Statement

In this work, we propose an effective solution to a technical problem: estimating and controlling for multiaccuracy and multicalibration using proxies. While proxies can be controversial and pose risks—such as reinforcing discrimination or compromising privacy—predictive models often learn sensitive attributes indirectly, leading to unintended harm. When used carefully, proxies can help mitigate these risks, as we demonstrate in this work. However, deploying proxies in real-world scenarios requires a careful evaluation of trade-offs through discussions with policymakers, domain experts, and other stakeholders. Ultimately, proxies should be employed responsibly and solely for assessing and promoting fairness.

## Acknowledgements

This work was supported by NIH award R01CA287422.

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

# A. Additional Theoretical Results

Here we present a version of Theorem 5.1 for multiaccuracy.

**Theorem A.1.** *Fix a distribution $\mathcal{D}$, initial model $f$, and set of proxy groups $\hat{\mathcal{G}}$. If a model $\hat{f}$ satisfies*

$$\mathsf{AE}^{max}(\hat{f}, \hat{\mathcal{G}}) < \min_{\hat{g} \in \hat{\mathcal{G}}} \mathsf{AE}_{\mathcal{D}}(f, \hat{g}) \tag{19}$$

$$\mathsf{MSE}(\hat{f}) \leq \mathsf{MSE}(f) \tag{20}$$

*then, it will have a smaller worst-case MA violation, i.e. $\beta(\hat{f}, \hat{\mathcal{G}}) \leq \beta(f, \hat{\mathcal{G}})$.*

A proof of this result is provided in Appendix B.3.

# B. Proofs

## B.1. Proof of Lemma 4.1

*Proof.* **Throughout these proofs, denote**

$$\mu_{i,j}^g = \mathbb{P}[g(X, Z) = i, \hat{g}(X) = j] \quad \text{and} \quad \mu_{i,j}^g(v) = \mathbb{P}[g(X, Z) = i, \hat{g}(X) = j \mid f(X) = v]. \tag{21}$$

We begin with proving the result for multiaccuracy.

*Step 1: Establishing the Multiaccuracy Bound*

Fix a distribution $\mathcal{D}$ and predictor $f$. Consider any group $g \in \mathcal{G}$ and its corresponding proxy $\hat{g} \in \hat{\mathcal{G}}$. Then,

$$\mathsf{AE}_{\mathcal{D}}(f, g) = \left| \mathbb{E}[g(X, Z)(f(X) - Y)] \right| \tag{22}$$

$$= \left| \mathbb{E}[g(X, Z)(f(X) - Y)] - \mathbb{E}[\hat{g}(X)(f(X) - Y)] + \mathbb{E}[\hat{g}(X)(f(X) - Y)] \right| \tag{23}$$

$$\leq \left| \mathbb{E}[g(X, Z)(f(X) - Y)] - \mathbb{E}[\hat{g}(X)(f(X) - Y)] \right| + \left| \mathbb{E}[\hat{g}(X)(f(X) - Y)] \right| \tag{24}$$

$$= \left| \mathbb{E}[g(X, Z)(f(X) - Y) - \hat{g}(X)(f(X) - Y)] \right| + \mathsf{AE}_{\mathcal{D}}(f, \hat{g}) \tag{25}$$

$$= \left| \mathbb{E}[(g(X, Z) - \hat{g}(X)) \cdot (f(X) - Y)] \right| + \mathsf{AE}_{\mathcal{D}}(f, \hat{g}) \tag{26}$$

$$\leq \mathbb{E}[|g(X, Z) - \hat{g}(X)| \cdot |f(X) - Y|] + \mathsf{AE}_{\mathcal{D}}(f, \hat{g}) \tag{27}$$

$$\leq \min\left( \sqrt{\mathbb{E}[|g(X, Z) - \hat{g}(X)|^2] \cdot \mathbb{E}[|f(X) - Y|^2]}, \ \mathbb{E}[|g(X, Z) - \hat{g}(X)|] \right) + \mathsf{AE}_{\mathcal{D}}(f, \hat{g}) \tag{28}$$

$$= \min\left( \sqrt{\mathsf{MSE}(f) \cdot \mathsf{err}(\hat{g})}, \ \mathsf{err}(\hat{g}) \right) + \mathsf{AE}_{\mathcal{D}}(f, \hat{g}). \tag{29}$$

Here we applied the triangle inequality in line 24, Jensen's inequality in line 27, and Holder's inequality in line 28.

*Step 2: Tightness of the Multiaccuracy Bound*

We now show these bounds are tight. To be precise, we will prove that there exists a joint distribution over the random variables $(f(X), Y, g(X, Z), \hat{g}(X))$ for which these bounds hold with equality.

Consider a group $g \in \mathcal{G}$ and its corresponding proxy $\hat{g} \in \hat{\mathcal{G}}$. First, consider the scenario where $\mathsf{MSE}(f) \leq \mathsf{err}(\hat{g})$ so that by the first result of Lemma 4.1 we have

$$\mathsf{AE}_{\mathcal{D}}(f, g) \leq \mathsf{AE}_{\mathcal{D}}(f, \hat{g}) + \sqrt{\mathsf{err}(\hat{g})} \cdot \sqrt{\mathsf{MSE}(f)}. \tag{30}$$

Consider the following data generating process:

- Conditioned on the event $\{g(X, Z) = \hat{g}(X)\}$, one has $f(X) = Y$,

- Conditioned on the event $\{g(X, Z) = 1, \hat{g}(X) = 0\}$, one has that $f(X) = \dfrac{\sqrt{\mathsf{MSE}(f)}}{\sqrt{\mathsf{err}(\hat{g})}}$ and $Y = 0$,

- $\mu^g_{0,1} = 0$ so that $\mathsf{err}(\hat{g}) = \mu^g_{1,0}$.

Then,

$$\mathbb{E}[g(X, Z)(f(X) - Y)] = \mathbb{E}[f(X) - Y \mid g(X, Z) = 1, \hat{g}(X) = 1]\mu^g_{1,1} + \mathbb{E}[f(X) - Y \mid g(X, Z) = 1, \hat{g}(X) = 0]\mu^g_{1,0} \tag{31}$$

$$= \sqrt{\mathsf{err}(\hat{g})} \cdot \sqrt{\mathsf{MSE}(f)}. \tag{32}$$

Further,

$$\mathbb{E}[\hat{g}(X)(f(X) - Y)] = \mathbb{E}[f(X) - Y \mid g(X, Z) = 1, \hat{g}(X) = 1]\mu^g_{1,1} + \mathbb{E}[f(X) - Y \mid g(X, Z) = 0, \hat{g}(X) = 1]\mu^g_{0,1} \tag{33}$$

$$= 0. \tag{34}$$

As a result,

$$\left| \mathbb{E}[g(X, Z)(f(X) - Y)] \right| = \left| \mathbb{E}[\hat{g}(X)(f(X) - Y)] + \sqrt{\mathsf{err}(\hat{g})} \cdot \sqrt{\mathsf{MSE}(f)} \right| \tag{35}$$

$$= \left| \mathbb{E}[\hat{g}(X)(f(X) - Y)] \right| + \sqrt{\mathsf{err}(\hat{g})} \cdot \sqrt{\mathsf{MSE}(f)}. \tag{36}$$

Now, consider the scenario where $\mathsf{MSE}(f) > \mathsf{err}(\hat{g})$ so that by the first result of Lemma 4.1 we have

$$\mathsf{AE}_{\mathcal{D}}(f, g) \leq \mathsf{AE}_{\mathcal{D}}(f, \hat{g}) + \mathsf{err}(\hat{g}). \tag{37}$$

Consider the following data generating process:

- Conditional on the event $\{g(X, Z) = \hat{g}(X)\}$, $f(X) \geq Y$

- Conditional on the event $\{g(X, Z) = 1, \hat{g}(X) = 0\}$, $f(X) = 1$ and $Y = 0$.

- $\mu^g_{0,1} = 0$ so that $\mathsf{err}(\hat{g}) = \mu^g_{1,0}$

Then,

$$\mathbb{E}[g(X, Z)(f(X) - Y)] = \mathbb{E}[f(X) - Y \mid g(X, Z) = 1, \hat{g}(X) = 1]\mu^g_{1,1} + \mathbb{E}[f(X) - Y \mid g(X, Z) = 1, \hat{g}(X) = 0]\mu^g_{1,0} \tag{38}$$

$$= \mathbb{E}[f(X) - Y \mid g(X, Z) = 1, \hat{g}(X) = 1]\mu^g_{1,1} + \mathsf{err}(\hat{g}). \tag{39}$$

Further,

$$\mathbb{E}[\hat{g}(X)(f(X) - Y)] = \mathbb{E}[f(X) - Y \mid g(X, Z) = 1, \hat{g}(X) = 1]\mu^g_{1,1} + \mathbb{E}[f(X) - Y \mid g(X, Z) = 0, \hat{g}(X) = 1]\mu^g_{0,1} \tag{40}$$

$$= \mathbb{E}[f(X) - Y \mid g(X, Z) = 1, \hat{g}(X) = 1]\mu^g_{1,1} \geq 0. \tag{41}$$

In passing, note that requiring $f(X) \geq Y$ above is much more than needed, but we take this for simplicity. Moving on, and as a result,

$$\left| \mathbb{E}[g(X, Z)(f(X) - Y)] \right| = \left| \mathbb{E}[\hat{g}(X)(f(X) - Y)] + \mathsf{err}(\hat{g}) \right| = \left| \mathbb{E}[\hat{g}(X)(f(X) - Y)] \right| + \mathsf{err}(\hat{g}). \tag{42}$$

We now prove the result for multicalibration.

*Step 1: Establishing the Multicalibration Bound*

Fix a distribution $\mathcal{D}$ and predictor $f$. Consider any group $g \in \mathcal{G}$ and its corresponding proxy $\hat{g} \in \hat{\mathcal{G}}$. Then,

$$\mathsf{ECE}_{\mathcal{D}}(f, g) = \mathbb{E}\left[\left|\mathbb{E}[g(X, Z)(f(X) - Y)|f(X) = v]\right|\right] \tag{43}$$

$$= \mathbb{E}\left[\left|\mathbb{E}[g(X, Z)(f(X) - Y)|f(X) = v]\right| - \left|\mathbb{E}[\hat{g}(X)(f(X) - Y)|f(X) = v]\right|\right. \tag{44}$$

$$\left. + \left|\mathbb{E}[\hat{g}(X)(f(X) - Y)|f(X) = v]\right|\right] \tag{45}$$

$$= \mathbb{E}\left[\left|\mathbb{E}[g(X, Z)(f(X) - Y)|f(X) = v]\right| - \left|\mathbb{E}[\hat{g}(X)(f(X) - Y)|f(X) = v]\right|\right] + \mathsf{ECE}_{\mathcal{D}}(f, \hat{g}) \tag{46}$$

$$\leq \mathbb{E}\left[\left|\mathbb{E}[g(X, Z)(f(X) - Y)|f(X) = v] - \mathbb{E}[\hat{g}(X)(f(X) - Y)|f(X) = v]\right|\right] + \mathsf{ECE}_{\mathcal{D}}(f, \hat{g}) \tag{47}$$

$$= \mathbb{E}\left[\left|\mathbb{E}[g(X, Z)(f(X) - Y) - \hat{g}(X)(f(X) - Y)|f(X) = v]\right|\right] + \mathsf{ECE}_{\mathcal{D}}(f, \hat{g}) \tag{48}$$

$$\leq \mathbb{E}\left[\mathbb{E}\left[\left|g(X, Z)(f(X) - Y) - \hat{g}(X)(f(X) - Y)\right| \middle| f(X) = v\right]\right] + \mathsf{ECE}_{\mathcal{D}}(f, \hat{g}) \tag{49}$$

$$= \mathbb{E}\left[\left|g(X, Z)(f(X) - Y) - \hat{g}(X)(f(X) - Y)\right|\right] + \mathsf{ECE}_{\mathcal{D}}(f, \hat{g}) \tag{50}$$

$$= \mathbb{E}\left[\left|(g(X, Z) - \hat{g}(X)) \cdot (f(X) - Y)\right|\right] + \mathsf{ECE}_{\mathcal{D}}(f, \hat{g}) \tag{51}$$

$$\leq \min\left(\sqrt{\mathbb{E}[|g(X, Z) - \hat{g}(X)|^2] \cdot \mathbb{E}[|f(X) - Y|^2]}, \ \mathbb{E}[|g(X, Z) - \hat{g}(X)|]\right) + \mathsf{ECE}_{\mathcal{D}}(f, \hat{g}) \tag{52}$$

$$= \min\left(\sqrt{\mathsf{MSE}(f) \cdot \mathsf{err}(g, \hat{g})}, \ \mathsf{err}(\hat{g})\right) + \mathsf{ECE}_{\mathcal{D}}(f, \hat{g}) \tag{53}$$

Here we applied the reverse triangle inequality in (47), Jensen's inequality in line (49), and Holder's inequality in line (52).

*Step 2: Tightness of the Multicalibration Bound*

We now show these bounds are tight. To be precise, we will prove that there exists a joint distribution over the random variables $(f(X), Y, g(X, Z), \hat{g}(X))$ for which these bounds hold with equality.

Consider a group $g \in \mathcal{G}$ and its corresponding proxy $\hat{g} \in \hat{\mathcal{G}}$. First, consider the scenario where $\mathsf{MSE}(f) \leq \mathsf{err}(\hat{g})$ so that by the first result of Lemma 4.1 we have

$$\mathsf{ECE}_{\mathcal{D}}(f, g) \leq \mathsf{ECE}_{\mathcal{D}}(f, \hat{g}) + \sqrt{\mathsf{err}(\hat{g})} \cdot \sqrt{\mathsf{MSE}(f)}. \tag{54}$$

Consider the same data generating process used to establish the MA bound, $\mathsf{AE}_{\mathcal{D}}(f, g) \leq \mathsf{AE}_{\mathcal{D}}(f, \hat{g}) + \sqrt{\mathsf{err}(\hat{g})} \cdot \sqrt{\mathsf{MSE}(f)}$. Then,

$$\mathbb{E}_{v \sim \mathcal{D}_f}\left|\mathbb{E}[g(X, Z)(f(X) - Y)|f(X) = v]\right| = \mathbb{E}_{v \sim \mathcal{D}_f}\left|\mathbb{E}[f(X) - Y \mid f(X) = v, g(X, Z) = 1, \hat{g}(X) = 1] \cdot \mu_{1,1}^g(v)\right. \tag{55}$$

$$\left. + \mathbb{E}[f(X) - Y \mid f(X) = v, g(X, Z) = 1, \hat{g}(X) = 0] \cdot \mu_{1,0}^g(v)\right| \tag{56}$$

$$= \mathbb{E}_{v \sim \mathcal{D}_f}\left|\frac{\sqrt{\mathsf{MSE}(f)}}{\sqrt{\mathsf{err}(\hat{g})}} \cdot \mu_{1,0}^g(v)\right| \tag{57}$$

$$= \frac{\sqrt{\mathsf{MSE}(f)}}{\sqrt{\mathsf{err}(\hat{g})}} \cdot \mathbb{E}_{v \sim \mathcal{D}_f}\left|\mu_{1,0}^g(v)\right| \tag{58}$$

$$= \frac{\sqrt{\mathsf{MSE}(f)}}{\sqrt{\mathsf{err}(\hat{g})}} \cdot \mu_{1,0}^g = \sqrt{\mathsf{MSE}(f)}\sqrt{\mathsf{err}(\hat{g})}. \tag{59}$$

Further,

$$\underset{v \sim \mathcal{D}_f}{\mathbb{E}} \left| \mathbb{E}[\hat{g}(X)(f(X) - Y)|f(X) = v] \right| = \underset{v \sim \mathcal{D}_f}{\mathbb{E}} \left| \mathbb{E}[f(X) - Y \mid f(X) = v, g(X, Z) = 1, \hat{g}(X) = 1] \cdot \mu_{1,1}^g(v) \right. \tag{60}$$

$$\left. + \mathbb{E}[f(X) - Y \mid f(X) = v, g(X, Z) = 0, \hat{g}(X) = 1] \cdot \mu_{0,1}^g(v) \right| \tag{61}$$

$$= 0. \tag{62}$$

As a result,

$$\underset{v \sim \mathcal{D}_f}{\mathbb{E}} \left| \mathbb{E}[g(X, Z)(f(X) - Y)|f(X) = v] \right| = \underset{v \sim \mathcal{D}_f}{\mathbb{E}} \left| \mathbb{E}[\hat{g}(X)(f(X) - Y)|f(X) = v] \right| + \sqrt{\mathsf{MSE}(f)}\sqrt{\mathsf{err}(\hat{g})}. \tag{63}$$

Now, consider the scenario where $\mathsf{MSE}(f) > \mathsf{err}(\hat{g})$ so that by the first result of Lemma 4.1 we have

$$\mathsf{ECE}_{\mathcal{D}}(f, g) \leq \mathsf{ECE}_{\mathcal{D}}(f, \hat{g}) + \mathsf{err}(\hat{g}). \tag{64}$$

Consider the same data generating process used to establish the MA bound, $\mathsf{AE}_{\mathcal{D}}(f, g) \leq \mathsf{AE}_{\mathcal{D}}(f, \hat{g}) + \mathsf{err}(\hat{g})$. Then,

$$\underset{v \sim \mathcal{D}_f}{\mathbb{E}} \left| \mathbb{E}[g(X, Z)(f(X) - Y)|f(X) = v] \right| = \underset{v \sim \mathcal{D}_f}{\mathbb{E}} \left| \mathbb{E}[f(X) - Y \mid f(X) = v, g(X, Z) = 1, \hat{g}(X) = 1] \cdot \mu_{1,1}^g(v) \right. \tag{65}$$

$$\left. + \mathbb{E}[f(X) - Y \mid f(X) = v, g(X, Z) = 1, \hat{g}(X) = 0] \cdot \mu_{1,0}^g(v) \right| \tag{66}$$

$$= \underset{v \sim \mathcal{D}_f}{\mathbb{E}} \left| \mathbb{E}[f(X) - Y \mid f(X) = v, g(X, Z) = 1, \hat{g}(X) = 1] \cdot \mu_{1,1}^g(v) + \mu_{1,0}^g(v) \right| \tag{67}$$

$$= \underset{v \sim \mathcal{D}_f}{\mathbb{E}} \left| \mathbb{E}[f(X) - Y \mid f(X) = v, g(X, Z) = 1, \hat{g}(X) = 1] \cdot \mu_{1,1}^g(v) \right| \tag{68}$$

$$+ \underset{v \sim \mathcal{D}_f}{\mathbb{E}}[\mu_{1,0}^g(v)] \tag{69}$$

$$= \underset{v \sim \mathcal{D}_f}{\mathbb{E}} \left| \mathbb{E}[f(X) - Y \mid f(X) = v, g(X, Z) = 1, \hat{g}(X) = 1] \cdot \mu_{1,1}^g(v) \right| + \mu_{1,0}^g. \tag{70}$$

Further,

$$\underset{v \sim \mathcal{D}_f}{\mathbb{E}} \left| \mathbb{E}[\hat{g}(X)(f(X) - Y)|f(X) = v] \right| = \underset{v \sim \mathcal{D}_f}{\mathbb{E}} \left| \mathbb{E}[f(X) - Y \mid f(X) = v, g(X, Z) = 1, \hat{g}(X) = 1] \cdot \mu_{1,1}^g(v) \right. \tag{71}$$

$$\left. + \mathbb{E}[f(X) - Y \mid f(X) = v, g(X, Z) = 0, \hat{g}(X) = 1] \cdot \mu_{0,1}^g(v) \right| \tag{72}$$

$$= \underset{v \sim \mathcal{D}_f}{\mathbb{E}} \left| \mathbb{E}[f(X) - Y \mid f(X) = v, g(X, Z) = 1, \hat{g}(X) = 1] \cdot \mu_{1,1}^g(v) \right| \tag{73}$$

As a result,

$$\underset{v \sim \mathcal{D}_f}{\mathbb{E}} \left| \mathbb{E}[g(X, Z)(f(X) - Y)|f(X) = v] \right| = \underset{v \sim \mathcal{D}_f}{\mathbb{E}} \left| \mathbb{E}[\hat{g}(X)(f(X) - Y)|f(X) = v] \right| + \mathsf{err}(\hat{g}). \tag{74}$$

The following bounds are tight in that we show there exists a distribution for which the bound is attained. The bounds can nonetheless be improved by assuming complete access to the marginal distributions over $(f(X), Y, \hat{g}(X))$ and $(g(X, Z), \hat{g}(X))$. In this case, one could derive tight bounds that agree with this observed information with techniques used in (Kallus et al., 2022; Bharti et al., 2024) that rely on the Fréchet inequalities. $\qquad \square$

### B.2. Proof of Theorem 4.2

*Proof.* We prove the result for multiaccuracy. Recall Lemma 4.1, which states that for any group $g$ and its proxy $\hat{g}$

$$\mathsf{AE}_{\mathcal{D}}(f, g) \leq \mathsf{F}(f, \hat{g}) + \mathsf{AE}_{\mathcal{D}}(f, \hat{g}) \tag{75}$$

Thus, $\forall g \in \mathcal{G}$,

$$\mathsf{AE}_{\mathcal{D}}(f, g) \leq \beta(f, \hat{\mathcal{G}}) := \max_{\hat{g} \in \hat{\mathcal{G}}} \mathsf{F}(f, \hat{g}) + \mathsf{AE}_{\mathcal{D}}(f, \hat{g}). \tag{76}$$

This proves that $f$ is $(\mathcal{G}, \beta(f, \hat{\mathcal{G}}))$-multiaccurate. The proof for multicalibration follows an identical argument. □

### B.3. Proof of Theorem 5.1

*Proof.* Fix a distribution $\mathcal{D}$, model $f$, set of groups $\mathcal{G}$ and its corresponding proxies $\hat{\mathcal{G}}$. Recall by Theorem 4.2 that $f$ is $(\mathcal{G}, \gamma(f,))$-multicalibrated where

$$\gamma(f, \hat{\mathcal{G}}) = \max_{\hat{g} \in \mathcal{G}} \min\left(\mathsf{err}(\hat{g}), \sqrt{\mathsf{MSE}(f) \cdot \mathsf{err}(\hat{g})}\right) + \mathsf{ECE}_{\mathcal{D}}(f, \hat{g}) \tag{77}$$

First, note that for $\mathsf{MSE}(f) > \mathsf{err}(\hat{g})$, the term

$$\min\left(\mathsf{err}(\hat{g}), \sqrt{\mathsf{MSE}(f) \cdot \mathsf{err}(\hat{g})}\right) \tag{78}$$

is constant with respect to $\mathsf{MSE}(f)$ and for $\mathsf{MSE}(f) \leq \mathsf{err}(\hat{g})$ it increases as $\mathsf{MSE}(f)$ increases.

Now, suppose another model $\hat{f}$ satisfies the following

$$\mathsf{ECE}^{\mathsf{max}}(\hat{f}, \hat{\mathcal{G}}) < \min_{\hat{g} \in \hat{\mathcal{G}}} \mathsf{ECE}_{\mathcal{D}}(f, \hat{g}) \tag{79}$$

$$\mathsf{MSE}(\hat{f}) \leq \mathsf{MSE}(f). \tag{80}$$

Then,

$$\gamma(\hat{f}, \hat{\mathcal{G}}) = \max_{\hat{g} \in \mathcal{G}} \min\left(\mathsf{err}(\hat{g}), \sqrt{\mathsf{MSE}(\hat{f}) \cdot \mathsf{err}(\hat{g})}\right) + \mathsf{ECE}_{\mathcal{D}}(\hat{f}, \hat{g}) \tag{81}$$

$$\leq \max_{\hat{g} \in \mathcal{G}} \min\left(\mathsf{err}(\hat{g}), \sqrt{\mathsf{MSE}(f) \cdot \mathsf{err}(\hat{g})}\right) + \mathsf{ECE}_{\mathcal{D}}(\hat{f}, \hat{g}) \tag{82}$$

$$\leq \max_{\hat{g} \in \mathcal{G}} \min\left(\mathsf{err}(\hat{g}), \sqrt{\mathsf{MSE}(f) \cdot \mathsf{err}(\hat{g})}\right) + \mathsf{ECE}^{\mathsf{max}}(\hat{f}, \hat{\mathcal{G}}) \tag{83}$$

$$\leq \max_{\hat{g} \in \mathcal{G}} \min\left(\mathsf{err}(\hat{g}), \sqrt{\mathsf{MSE}(f) \cdot \mathsf{err}(\hat{g})}\right) + \min_{\hat{g} \in \hat{\mathcal{G}}} \mathsf{ECE}_{\mathcal{D}}(f, \hat{g}) \tag{84}$$

$$\leq \max_{\hat{g} \in \mathcal{G}} \min\left(\mathsf{err}(\hat{g}), \sqrt{\mathsf{MSE}(f) \cdot \mathsf{err}(\hat{g})}\right) + \mathsf{ECE}_{\mathcal{D}}(f, \hat{g}) \tag{85}$$

$$= \gamma(f, \hat{\mathcal{G}}). \tag{86}$$

The proof of the multiaccuracy result in Theorem A.1 follows an identical argument. □

## C. Additional Experiment Details

### C.1. ACS Experiments

**Models.** In the ACSIncome and PubCov experiments, we use Random Forests as the proxy models $\hat{g}$, and train three types of models $f$: logistic regression, decision tree, and Random Forest.

**Results.** The sensitive groups we use along with their proxy errors are reported in Tables 4 and 5. All MA related results for the different models $f$ are presented in Appendices E.1 and E.2. Note, all of the models are MA with respect to $\mathcal{G}$ and $\hat{\mathcal{G}}$. As a result, adjusting with respect to the proxies provides no benefit. All MC related results for the different models $f$ are presented in Appendices E.4 and E.5. Note, the logistic regression and Random Forest models are highly multicalibrated with respect to $\mathcal{G}$ and $\hat{\mathcal{G}}$. As a result, adjusting provides no benefit. On the other hand, the decision tree is grossly uncalibrated with respect to some proxies. As a result, we see a benefit in multicalibrating with respect to the proxies.

### C.2. CheXpert

**Models.** In the CheXpert experiment, we follow Glocker et al. (2023) and train a DenseNet-121 model for the to predict `race` and `sex`. For the models $f$, we use three types. The first is a decision tree classifier trained on features extracted from a DenseNet-121 model (Huang et al., 2017) pretrained on ImageNet (Deng et al., 2009). The second is a linear model (Breiman, 2001) trained on the same features. The third is a DenseNet-121 model trained end-to-end on the raw X-ray images.

**Results.** The groups used along with proxy errors are reported in Table 6. All multiaccuracy related results for the different types of models $f$ are presented in Appendix E.3. Note, all of the models are multiaccurate with respect to $\mathcal{G}$ and $\hat{\mathcal{G}}$. As a result, adjusting provides no benefit. All multicalibration related results for the different types of models $f$ are presented in Appendix E.6. Note, the logistic regression and fully trained DenseNet-121 models are highly multicalibrated with respect to $\mathcal{G}$ and $\hat{\mathcal{G}}$. As a result, adjusting provides no benefit. On the other hand, the decision tree is grossly uncalibrated with respect to some proxies. As a result, we see a benefit in multicalibrating with respect to the proxies.

## D. Additional Tables

| Group | err($\hat{g}$) |
|---|---|
| Black Adults | 0.044 |
| Black Women | 0.027 |
| Women | 0.000 |
| Never Married | 0.000 |
| American Indian | 0.007 |
| Seniors | 0.000 |
| White Women | 0.123 |
| Multiracial | 0.047 |
| White Children | 0.002 |
| Asian | 0.060 |

*Table 4.* Sensitive groups and the proxy errors used in the ACSIncome experiment

| Group | err($\hat{g}$) |
|---|---|
| Black Adults | 0.044 |
| Black Women | 0.027 |
| Women | 0.000 |
| Never Married | 0.000 |
| American Indian | 0.007 |
| White Women | 0.123 |
| Multiracial | 0.047 |
| White Children | 0.002 |
| Asian | 0.060 |

*Table 5.* Sensitive groups and the proxy errors used in the ACSPubCov experiment

| Group | err($\hat{g}$) |
|---|---|
| Men | 0.027 |
| Women | 0.027 |
| White | 0.920 |
| Asian | 0.068 |
| Black | 0.039 |
| Asian Men | 0.039 |
| Asian Women | 0.034 |
| Black Men | 0.021 |
| Black Women | 0.020 |
| White Men | 0.067 |
| White Women | 0.062 |

*Table 6.* Sensitive groups and the proxy errors used in the CheXpert experiment

# E. Additional Figures

## E.1. Multiaccuracy results for ACSIncome

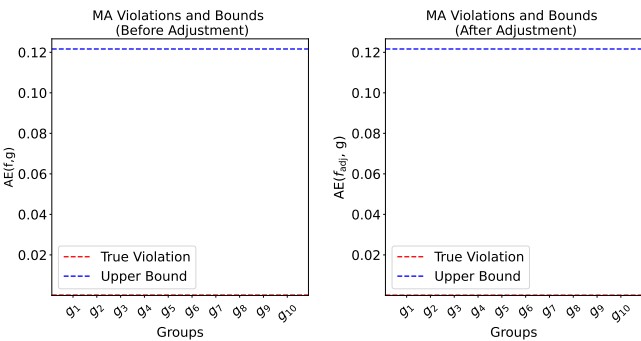

*Figure B.1.* $\mathsf{AE}(f, g)$, $\mathsf{AE}^{\max}(f, g)$ (dotted red line), and worst case violations (dotted blue line) of the original model $f$ and adjusted model $f_{\text{adj}}$ on ACSIncome. Here, $f$ is a logistic regression.

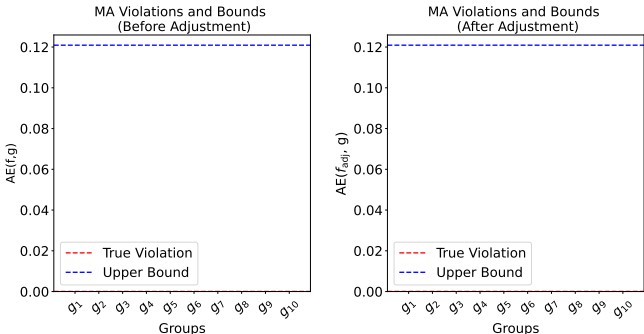

*Figure B.2.* $\mathsf{AE}(f, g)$, $\mathsf{AE}^{\max}(f, g)$ (dotted red line), and worst case violations (dotted blue line) of the original model $f$ and adjusted model $f_{\text{adj}}$ on ACSIncome. Here, $f$ is a decision tree.

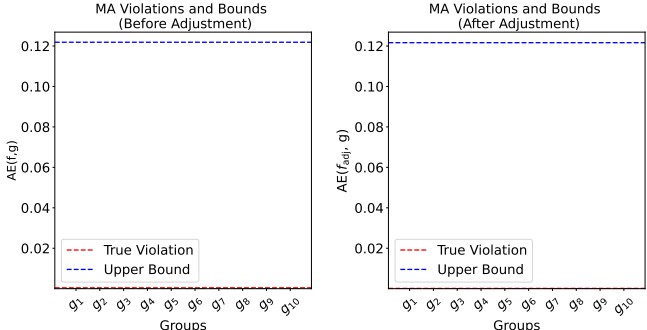

*Figure B.3.* $\mathsf{AE}(f, g)$, $\mathsf{AE}^{\max}(f, g)$ (dotted red line), and worst case violations (dotted blue line) of the original model $f$ and adjusted model $f_{\text{adj}}$ on ACSIncome. Here, $f$ is a Random Forest.

## E.2. Multiaccuracy results for ACSPubCov

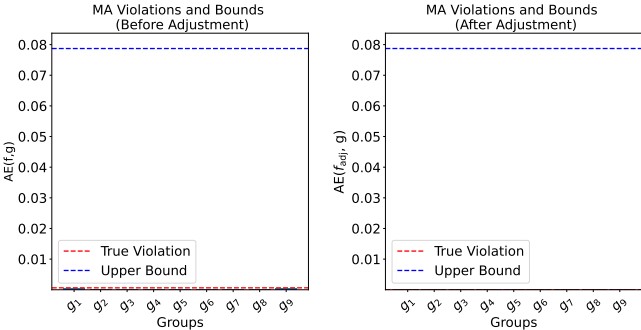

*Figure B.4.* $\mathsf{AE}(f, g)$, $\mathsf{AE}^{\mathsf{max}}(f, g)$ (dotted red line), and worst case violations (dotted blue line) of the original model $f$ and adjusted model $f_{\mathrm{adj}}$ on ACSPubCov. Here, $f$ is a logistic regression.

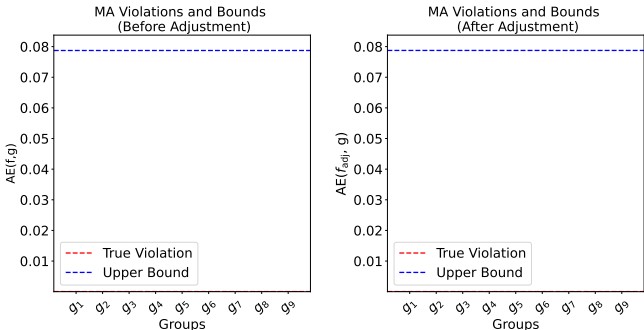

*Figure B.5.* $\mathsf{AE}(f, g)$, $\mathsf{AE}^{\mathsf{max}}(f, g)$ (dotted red line), and worst case violations (dotted blue line) of the original model $f$ and adjusted model $f_{\mathrm{adj}}$ on ACSPubCov. Here, $f$ is a decision tree.

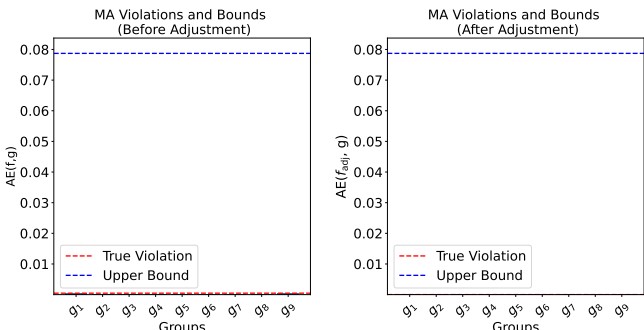

*Figure B.6.* $\mathsf{AE}(f, g)$, $\mathsf{AE}^{\mathsf{max}}(f, g)$ (dotted red line), and worst case violations (dotted blue line) of the original model $f$ and adjusted model $f_{\mathrm{adj}}$ on ACSPubCov. Here, $f$ is a Random Forest.

### E.3. Multiaccuracy results for CheXpert

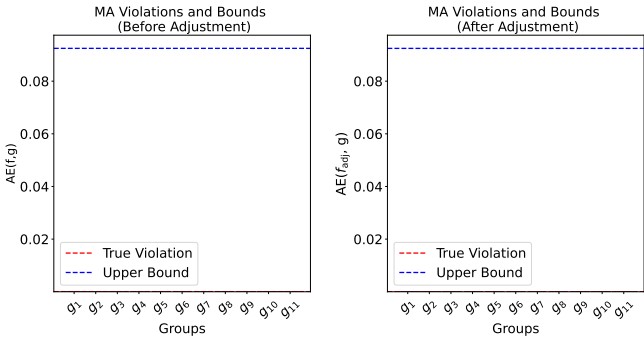

*Figure B.7.* $\mathsf{AE}(f, g)$, $\mathsf{AE}^{\mathsf{max}}(f, g)$ (dotted red line), and worst case violations (dotted blue line) of the original model $f$ and adjusted model $f_{\text{adj}}$ on CheXpert. Here, $f$ is a logistic regression.

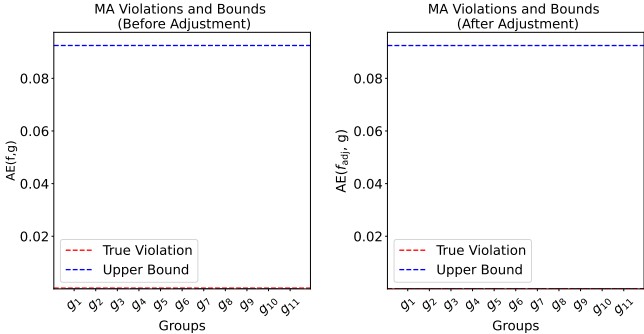

*Figure B.8.* $\mathsf{AE}(f, g)$, $\mathsf{AE}^{\mathsf{max}}(f, g)$ (dotted red line), and worst case violations (dotted blue line) of the original model $f$ and adjusted model $f_{\text{adj}}$ on CheXpert. Here, $f$ is a decision tree.

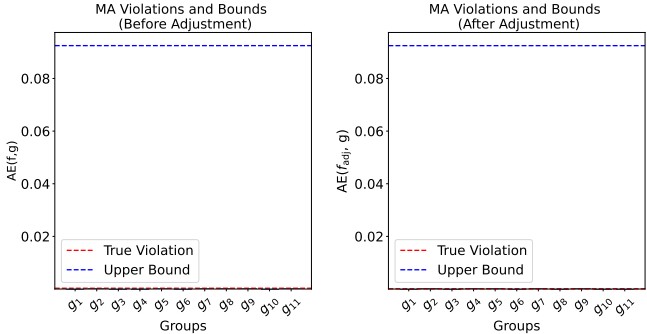

*Figure B.9.* $\mathsf{AE}(f, g)$, $\mathsf{AE}^{\mathsf{max}}(f, g)$ (dotted red line), and worst case violations (dotted blue line) of the original model $f$ and adjusted model $f_{\text{adj}}$ on CheXpert. Here, $f$ is a DenseNet-121 model.

## E.4. Multicalibration results for ACSIncome

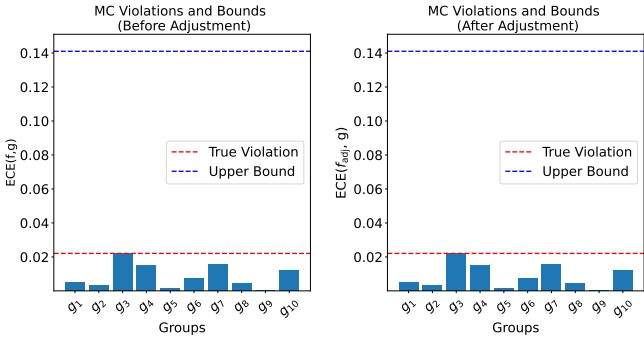

*Figure B.10.* ECE, ECEmax (dotted red line), and worst case violations (dotted blue line) of the original model $f$ and adjusted model $f_{\text{adj}}$ on ACSIncome. Here, $f$ is a logistic regression.

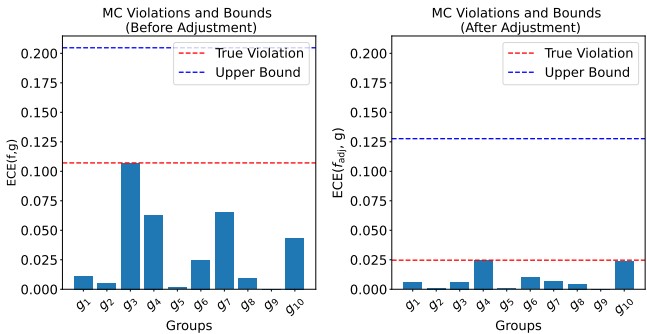

*Figure B.11.* ECE, ECEmax (dotted red line), and worst case violations (dotted blue line) of the original model $f$ and adjusted model $f_{\text{adj}}$ on ACSIncome. Here, $f$ is a decision tree.

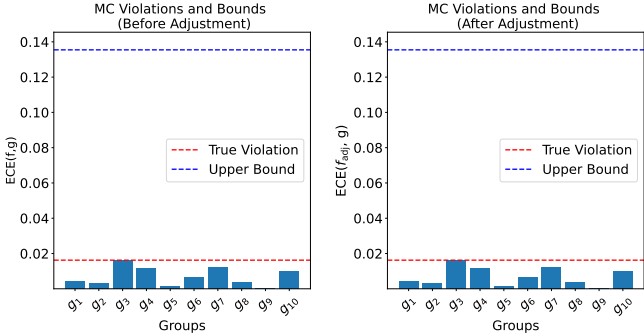

*Figure B.12.* ECE, ECEmax (dotted red line), and worst case violations (dotted blue line) of the original model $f$ and adjusted model $f_{\text{adj}}$ on ACSIncome. Here, $f$ is a Random Forest.

## E.5. Multicalibration results for ACSPubCov

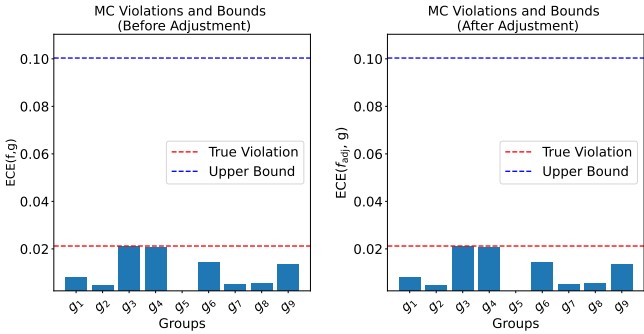

*Figure B.13.* ECE, ECE$^{\text{max}}$ (dotted red line), and worst case violations (dotted blue line) of the original model $f$ and adjusted model $f_{\text{adj}}$ on ACSPubCov. Here, $f$ is a logistic regression.

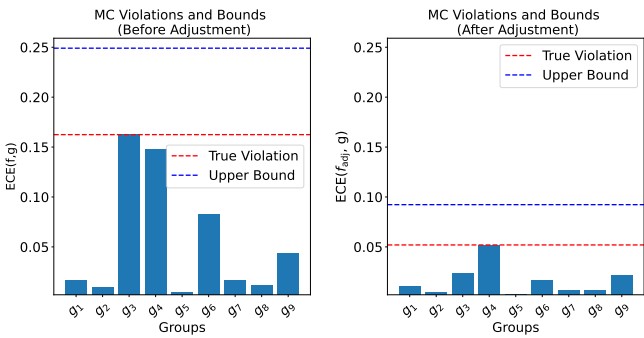

*Figure B.14.* ECE, ECE$^{\text{max}}$ (dotted red line), and worst case violations (dotted blue line) of the original model $f$ and adjusted model $f_{\text{adj}}$ on ACSPubCov. Here, $f$ is a decision tree.

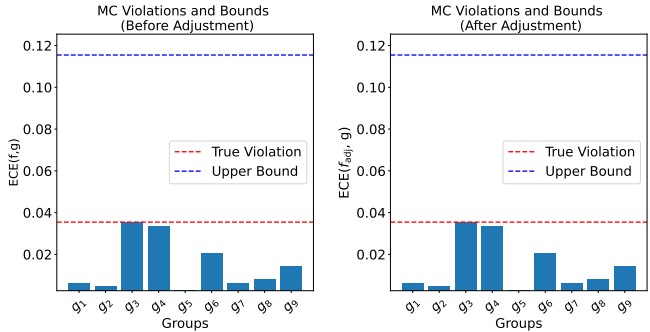

*Figure B.15.* ECE, ECE$^{\text{max}}$ (dotted red line), and worst case violations (dotted blue line) of the original model $f$ and adjusted model $f_{\text{adj}}$ on ACSPubCov. Here, $f$ is a Random Forest.

### E.6. Multicalibration results for CheXpert

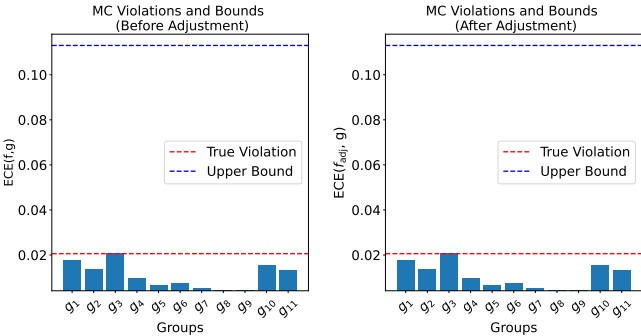

*Figure B.16.* ECE, ECE^max (dotted red line), and worst case violations (dotted blue line) of the original model $f$ and adjusted model $f_{\text{adj}}$ on CheXpert. Here, $f$ is a logistic regression.

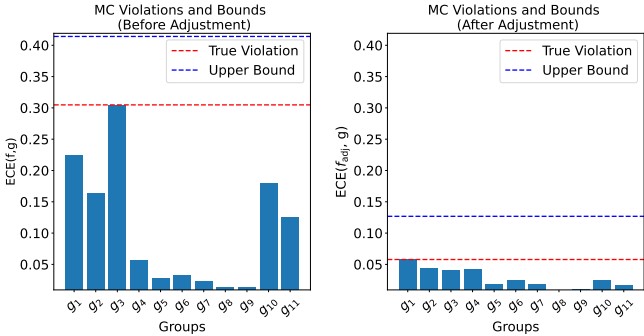

*Figure B.17.* ECE, ECE^max (dotted red line), and worst case violations (dotted blue line) of the original model $f$ and adjusted model $f_{\text{adj}}$ on CheXpert. Here, $f$ is a decision tree.

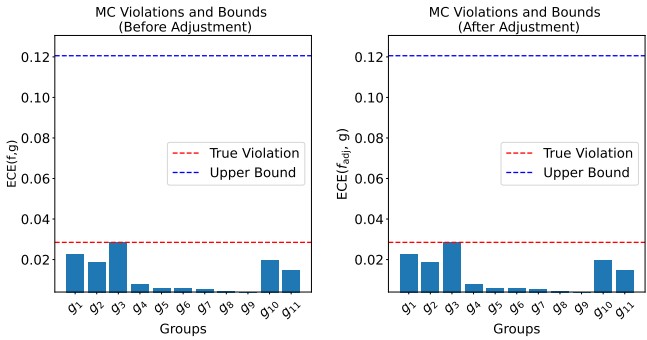

*Figure B.18.* ECE, ECE^max (dotted red line), and worst case violations (dotted blue line) of the original model $f$ and adjusted model $f_{\text{adj}}$ on CheXpert. Here, $f$ is a DenseNet-121 model.

