# OpenReview forum: "Multiaccuracy and Multicalibration via Proxy Groups"
_ICML.cc/2025/Conference — ICML 2025 poster_

### Official Review · Reviewer_hLPe · 2025-03-10

**Overall Recommendation:** 2

**Summary:**

The authors propose way to measure multicalibration/multiaccuracy by leveraging proxies (that way is to necessary to have access to the data) in order to compute worst case scenarios (formalized as an upper bound on their proposed metrics). They show that postprocessing model to satisfy multicalibration and multi-accuracy across proxies ends up in a reduction of the worst case (the computed upper bounds) violations.

They prove in theorem 4.2 how the multi-calibration/multi-accuracy of a model with respect to G is upper bounded with the multi-calibration/multi-accuracy with respect to the proxy.

The experimental results back up the developed theory .

Weaknesses:
- The last result, how the standard post-processing for multicalibration on proxy groups reduces the worst case could be a corollary from the main result (theorem 4.2). Is not clear to me why to devote a whole section to introduce standard procedures.
- For me there is a mismatch on the motivation the authors used for using proxy-groups and their methods. They claim using population level values allows for better privacy (which I agree) but then they present a method which still needs individual level data about the proxy groups. Moreover, given that the final algorithm is the standard postprocessig to achieve multicalibration, the only real information that is being omitted are which ones were the original groups of interests.

Strengths:
- The main result is interesting and timely. The experiments are well thought for backing their claims and usefulness of their approach of using proxy groups.
- The method is dependent both on the soundness of the proxies and the quality of the model.

Overall I think the idea is good but the paper has very superficial analysis, again section five is superfluous. Perhaps will be better to study further the relationship between the quality of the model vs the quality of the groups, obtaining thus a result akin to the optimal bound that the practitioner can use as a reference for when to improve their proxies or when to improve their model.

Finally, although the experiments are sound, is not clear to me how they improve fairness. I would like to see frorm a decision making point of view, how the proposed method would be used in real life.

**Claims And Evidence:**

The theory results are well presented, correctly proved and backed by relevant experiments.

**Essential References Not Discussed:**

Not to my knowledge

**Experimental Designs Or Analyses:**

The experiments are well thought and are relevant for the paper.

**Methods And Evaluation Criteria:**

They do.

**Other Comments Or Suggestions:**

...See weakness in the summary section

**Other Strengths And Weaknesses:**

See weakness in the summary section

**Questions For Authors:**

- Is there a way to know when to refined the proxies versus when is more beneficial to instead improve the accuracy of the model?

**Relation To Broader Scientific Literature:**

See weakness in the summary section

**Theoretical Claims:**

Th proofs are correct.

---

> ### Author Rebuttal · Authors · 2025-03-31
>
> Thank you for the review!
>
> **Concern 1: Motivation and Method Mismatch**
>
> We believe there might be a misunderstanding of the motivation behind our work - we apologize if it was not sufficiently clear.
>
> Our goal is to develop a predictor $f$: $\mathcal{X} \rightarrow [0,1]$ that is multiaccurate/multicalibrated across a set of sensitive groups $\mathcal{G} = \set{g: \mathcal{X} \times \mathcal{Z}: \rightarrow \set{0,1}}$. However, we do not observe sensitive information $Z$ and instead have access to the marginal distribution over features $X$ and labels $Y$, denoted as $\mathcal{D}_{\mathcal{X}\mathcal{Y}}$. Thus, note that once cannot evaluate the true MA/MC violations with respect to the true grouping functions $g \in \mathcal{G}$ since they are functions of unobserved $Z$. This is a common and important problem often seen in healthcare and government applications [1,2].
>
> Thus, we employ the popular approach of using proxies [3,4], denoted as $\hat{g}: \mathcal{X} \rightarrow \set{0,1}$, of the true groups $g$. We show that with proxies (and their misclassification rates), while we can’t measure the MA/MC violations over the true groups, we can provide upper bounds that certify that their worst-case violations are below a given level. Moreover, we show that the worst-case bound is actionable; i.e. it can be reduced by making $f$ multiaccurate/multicalibrated with respect to the groups defined by the proxies. This is useful because:
>
>  1) If we can ensure that the worst-case violation  $< \alpha$, then the true violation (across the true, unobserved group) is $< \alpha$ as well!
>
> 2) We do not need to develop a new algorithm, and instead can leverage existing ones that generate models that are multiaccurate/multicalibrated for any arbitrary set of groups.
>
> In conclusion, one can get *approximate* MA/MC guarantees via reducing worst-case violations.
>
> We are not trying to “claim that using population values allows for better privacy”, and apologize if this was not clear.
>
> Additionally, the reviewer asks, “They present a method which still needs individual-level data about the proxy groups?”. Note that the algorithms take as input any set of groups and, as currently presented, there are multiple instances of $g(X, Z)$ written. We see how this could be confusing, as it implies that we need access to $Z$. However, we want to clarify that when using the proxies, all instances of $g(X, Z)$ are replaced by $\hat{g}(X)$. As a result, with the proxies as inputs to the algorithms, all sensitive information $Z$ is naturally omitted.
>
> **Concern 2: Need for section 5**
>
> We kindly disagree that section 5 is superficial.  For many readers not familiar with the literature, it may not be immediately clear from Theorem 4.2 how post-processing for multicalibration on proxy groups would reduce the worst-case violation, and to what level $\alpha$ one needs to multicalibrate. Our theorem makes this clear. Importantly, Section 5 shows that enforcing MA/MC does not degrade the MSE, which we need to ensure that our bounds get smaller when enforcing MA/MC across the proxies.
>
> **Additional Questions/Concerns**
>
> *Although the experiments are sound, is not clear to me how they improve fairness. I would like to see from a decision-making point of view, how the proposed method would be used in real life*
>
> The notions of fairness we are focused on are multiaccuracy/multicalibration [5]. Our theory and experiments clearly show that these notions of fairness (their worst-case violations) can be provably reduced. We believe this is very useful: Suppose one needs to build a model that is $\alpha$-MC across the true groups. With our results, we can reduce the worst-case violation and, if it is less than $\alpha$, then we know the true violation is less than $\alpha$ as well - without requiring knowledge of the true groups!
>
> *Is there a way to know when to refine the proxies versus when it is more beneficial to instead improve the accuracy of the model?*
>
> By "beneficial", does the Rev. mean reducing the worst-case bounds? Based on our results, neither is more beneficial than the other since improving the proxies or improving the accuracy of $f$ will always reduce the bound. Additionally, multicalibrating with respect to the proxies **will** reduce the bound.
>
> Thank you for your questions and comments. We hope we have addressed them in a clear and satisfactory manner, and we'll be happy to address any outstanding questions!
>
> [1] "Advancing healthcare equity through improved data collection", J.S. Weissman et al. New England Journal of Medicine, 2011.
> [2] "Improving fairness in machine learning systems: What do industry practitioners need?", K. Holstein et al. CHI 2019.
> [3] "Using Bayesian imputation to assess racial and ethnic disparities in pediatric performance measures", D. Brown et al. Health services research 2016.
> [5] "Calibration for the (Computationally-Identifiable) Masses", Úrsula Hébert-Johnson et al. ICML 2018.

---

### Official Review · Reviewer_VC5A · 2025-03-11

**Overall Recommendation:** 4

**Summary:**

The paper "Multiaccuracy and Multicalibration via Proxy Groups" addresses the challenge of ensuring fairness in predictive machine learning models when sensitive group data is missing or incomplete. The authors focus on two fairness notions—multiaccuracy and multicalibration—which aim to ensure that model predictions are unbiased and well-calibrated across groups.
The paper demonstrates that proxy-sensitive attributes (features correlated with true sensitive attributes) can be used to derive actionable upper bounds on the true multiaccuracy and multicalibration violations. This allows practitioners to assess worst-case fairness violations even when true sensitive group data is unavailable.
The authors show that enforcing multiaccuracy and multicalibration using proxy-sensitive attributes can significantly mitigate fairness violations for the true (but unknown) demographic groups. They introduce computational methods to adjust models to satisfy these fairness criteria.
Finally, the study evaluates the proposed methods on multiple datasets, including ACSIncome, ACSPublicCoverage, and CheXpert (a medical imaging dataset). The results demonstrate that enforcing fairness across proxies leads to substantial reductions in worst-case fairness violations.

**Claims And Evidence:**

Some claims are well supported.
1) The authors derive provable upper bounds on multiaccuracy and multicalibration violations using proxy-sensitive attributes. These bounds are mathematically justified and clearly stated in Theorem 4.2 and Lemma 4.1. The theoretical results align with known fairness literature, and the derivations appear sound.
2) The authors present two algorithms (Multiaccuracy Regression and Multicalibration Boosting) that adjust models based on proxy attributes to reduce fairness violations. Theoretical results (Theorem 5.1 and Theorem 5.3) prove that these algorithms reduce worst-case violations while maintaining or improving predictive performance.
3) Empirical results on ACSIncome, ACSPublicCoverage, and CheXpert datasets demonstrate that the proposed methods successfully reduce fairness violations. Figures 1–4 illustrate reductions in worst-case multicalibration errors, showing that fairness guarantees improve after post-processing.
Nevertheless, some claims are still non-supported. For instance, the very last sentence of the introduction is "Even when sensitive information is incomplete or inaccessible, proxies can extend approximate multiaccuracy and multicalibration protections in a meaningful way." I did not see an explicit discussion on that point Also (see discussion below), "our methods offer, for the first time, the possibility to certify and correct multiaccuracy and multicalibration without requiring access to ground truth group data". I think that some related research on fairness with partially observed sensitive attributes suggests that bounding fairness violations using proxies has been explored before, such as
Awasthi, P., Kleindessner, M., and Morgenstern, J. Equalized odds postprocessing under imperfect group information. In International Conference on Artificial Intelligence and Statistics, pp. 1770–1780. PMLR, 2020.
or
Bharti, B., Yi, P., and Sulam, J. Estimating and controlling for equalized odds via sensitive attribute predictors. Advances in neural information processing systems, 36, 2024.
And finally, at the end of section 5, it is claimed that "Applying our methods ensures stronger fairness guarantees," but while post-processing reduces fairness violations with respect to proxies, it does not guarantee fairness with respect to the true sensitive attributes (which remain unobserved).

**Essential References Not Discussed:**

The definition of multicalibration used here seems to be weaker than that of the original work
Hebert-Johnson, U., Kim, M., Reingold, O., &amp; Rothblum, G. (2018). Multicalibration: Calibration for the (computationally-identifiable) masses. In J. Dy &amp; A. Krause (Eds.), Proceedings of the 35th International Conference on Machine Learning (Vol. 80, pp. 1939–1948).
It is important to explicitly clarify this distinction and its implications early in the paper, particularly the relationship between the definition based on the ECE and the definition (3.2) of multicalibration in Hebert-Johnson et al. (2018).
Additionally, the derived upper bounds should be explicitly linked to this weaker definition of multicalibration.

**Experimental Designs Or Analyses:**

I noticed the lack of a controlled synthetic dataset. There is no dataset where ground truth fairness violations are explicitly known. This makes it impossible to verify whether proxy-based fairness estimates accurately reflect reality. A synthetic dataset (where fairness violations are designed and known) could have served as a check.
Some proxy groups (e.g., “Women” in Table 1 with zero error) may be too accurate, suggesting that another feature (e.g., pregnancy status ?) strongly correlates with the sensitive attribute.

**Methods And Evaluation Criteria:**

The methods assume that proxy attributes are sufficiently good approximations of true sensitive attributes. However, the paper does not systematically test when proxies fail or lead to misleading fairness estimates. A stronger evaluation would vary proxy errors to test robustness.
The paper does not benchmark against other fairness estimation techniques that work with missing sensitive data (e.g., Bayesian imputation, adversarial debiasing). Adding comparisons would help assess whether proxy-based methods are superior, complementary, or limited in certain contexts.
Further, since fairness bounds depend on proxy quality and sample size, reporting confidence intervals or uncertainty estimates would make the results more robust.

**Other Comments Or Suggestions:**

see previous box

**Other Strengths And Weaknesses:**

The paper is very interesting.
The fairness bounds are derived under the assumption that proxy attributes adequately approximate true sensitive attributes, but this assumption is never explicitly validated. Furthermore, it does not explore how fairness guarantees degrade when proxies are inaccurate.
It does not benchmark against other methods for fairness estimation without sensitive attributes, such as Bayesian imputation for fairness estimation (as in Chen et al., 2019) or worst-case fairness estimation (as in Kallus et al., 2022)

**Questions For Authors:**

none

**Relation To Broader Scientific Literature:**

When sensitive attributes are missing or incomplete, fairness evaluation becomes difficult, and several approaches have been proposed. Important references are given.

**Theoretical Claims:**

Yes, all of them. The mathematics are rather straightforward

---

> ### Author Rebuttal · Authors · 2025-03-31
>
> Thank you for the review! We are glad you enjoyed the paper. Our responses to your questions and concerns are below.
>
> **Concern 1: Validity of some claims**
>
> We apologize if some claims seem unsupported; let us clarify:
>
> *Claim 1: "even when sensitive information is incomplete or inaccessible, proxies can extend approximate multiaccuracy and multicalibration protections in a meaningful way"*
>
> With proxies, we can establish non-trivial upper bounds on true MA/MC violations. Denote this bound as $B$. Then, while one cannot determine the exact violations, one can still provably assert that the model is $B$-multiaccurate/multicalibrated, providing a meaningful guarantee on the MA/MC violations of $f$. We will reword the statement to accurately reflect this point.
>
> *Claim 2: "our methods offer, for the first time, the possibility to certify and correct multiaccuracy and multicalibration without requiring access to ground truth group data"*
>
> You are correct that other works have also explored fairness by bounding fairness violations using proxies - as we cite in our manuscript. However, all of these works were concerned with parity- or group-based notions of fairness, such as demographic parity, equalized odds, equal opportunity, etc. Our work is the first to use proxies to bound MA/MC violations, which are quite different notions of fairness that are not parity-based [1].
>
> *Claim 3: "Applying our methods ensures stronger fairness guarantees"*
>
> We apologize that this claim suggests our methods allow us to control the true MA/MC violations. What we meant to convey is that our methods allow us to get stronger guarantees on the worst-case violations, which we believe (and demonstrated) can be practically useful. This is because, in our setting, the true violation cannot be evaluated. However, if one can modify $f$ such that our upper bound is less than $\alpha$, we can conclude that the true violation is also less than $\alpha$. Thus, our theory and methods are useful. We will make sure to clarify this point in the revised manuscript.
>
> **Concern 2: Potential failure of proxies**
>
> We believe there might be a slight misunderstanding here. In this work, we make no assumptions on how good or bad the proxies are. Additionally, our goal is not to assess how the true MA/MC guarantee changes as the proxies change. In the demographically scarce setting that we study, the true MA/MC violations cannot be identified/determined. Thus, we focus on providing bounds on these violations, which can be interpreted as worst-case violations. When analyzing the bounds (see e.g. Lemma 4.1), it's clear that if the proxies are bad, then our bounds will be naturally large, indicating the MA/MC violations could be large as well. Likewise, as the proxies become more accurate, our bound collapses to the true violation.
>
> Regardless of the quality of the proxies, our bounds always hold (i.e. we don't require any assumptions on them), and our methodology will always reduce the worst-case violation. Finally, the reason we do not benchmark against other methods is that our focus was not on studying *how* to build proxies for accurate fairness estimation. Instead, our focus is to study how to obtain practical MA/MC guarantees with *any set of proxies* that might be available via a general worst-case analysis.
>
> **Concern 3: questions about experiments**
>
> We agree that a synthetic experiment would be good! It would allow us to showcase when our bounds are tight (see rebuttal to reviewer LB2V) and analyze how the true fairness violations and proxy-fairness violations differ as a function of the proxy error. We plan to include a synthetic example in the revised manuscript. Thank you for the great suggestion!
>
> Additionally, you observe that some proxy groups (e.g., “Women” in Table 1 with zero error) may be too accurate, suggesting that another feature (e.g., pregnancy status ?) strongly correlates with the sensitive attribute. This is correct! This point comes to illustrate that in many real-world scenarios, even without observing $Z$, we can learn highly accurate proxies and use them to provide meaningful fairness estimates via worst-case analysis.
>
> **Additional comments**
>
> *Definition of Multicalibration*
>
> Thank you for pointing this out! You are correct in saying that the definition of multicalibration is different from the one proposed in [1]. The one we use is also referred to as "multicalibration" in other works [2,3] but, to be more precise, it should be referred to as approximate-multicalibration, or multicalibration in expectation. We will make this distinction very clear in the revised manuscript.
>
> [1] "Calibration for the (Computationally-Identifiable) Masses", Úrsula Hébert-Johnson et al. ICML 2018.
> [2] "Swap Agnostic Learning, or Characterizing Omniprediction via Multicalibration", Gopalan et al.. NeurIPS 2023.
> [3] "Multicalibrated Regression for Downstream Fairness", Globus-Harris et al. AIES 2023.

---

> > ### Comment · Reviewer_VC5A · 2025-04-08
> >
> > Thanks. I confirm my Overall Recommendation: 4: Accept

---

### Official Review · Reviewer_HMNE · 2025-03-14

**Overall Recommendation:** 4

**Summary:**

In this paper, the authors study the problem of fairness in ML. To be specific, they focus on a scenario where different groups are evaluated independently with respect to their accuracies and calibration errors (coined as Multiaccuracy and Multicalibration fairness in the literature). The literature has studied Multiaccuracy and Multicalibration in scenarios where sensitive attributes are available. The authors address this limitation by (i) providing theoretical bounds for Multiaccuracy and Multicalibration fairness if proxy attributes/groups are used and (ii) proposing/adapting algorithms to improve Multiaccuracy and Multicalibration fairness of ML algorithms without sensitive attribute information.

## update after rebuttal

I've read the comments by other reviewers and the rebuttal provided the authors. Therefore, I keep my original acceptance recommendation.

**Claims And Evidence:**

Yes.

**Essential References Not Discussed:**

None.

**Experimental Designs Or Analyses:**

Yes, all of them.

**Methods And Evaluation Criteria:**

Yes.

**Other Comments Or Suggestions:**

Minor comments:

- What is v in the ECE definition on Line 186?
- "data-scare regimes" => "data-scarce regimes".
- "notefirst that" => "note first that".

**Other Strengths And Weaknesses:**

Strengths:
+ Addressing bias and fairness in scenarios where sensitive attributes are unavailable is an important challenge in ML fairness.
+ Multiaccuracy and Multicalibration are important fairness definitions.
+ The paper provides theoretical supports as well as experimental results on several datasets.
+ Overall, the paper is very well written and easy to follow.

Weaknesses:

I am generally happy with the paper but I would like to state a few things:

1. I am generally unhappy with the proxy attribute approach relying on a separate model trained to estimate the sensitive attributes/groups. We are introducing a source of error into a very sensitive issue. If there turns out to be low fairness, it is not clear what the source of the problem is.

2. "the proxies exhibit some error, albeit small." => For completeness, please provide these figures.

3. Figs 1-4 do not have the labels for the axes and the lines. This makes it difficult to compare the subplots and evaluate the results.

**Questions For Authors:**

Please see Weaknesses.

**Relation To Broader Scientific Literature:**

The literature has studied Multiaccuracy and Multicalibration in scenarios where sensitive attributes are available. The authors address this limitation by (i) providing theoretical bounds for Multiaccuracy and Multicalibration fairness if proxy attributes/groups are used and (ii) proposing/adapting algorithms to improve Multiaccuracy and Multicalibration fairness of ML algorithms without sensitive attribute information.

**Theoretical Claims:**

Partially.

---

> ### Author Rebuttal · Authors · 2025-03-31
>
> Thank you for your careful review! We are glad you enjoyed the paper. Our responses to your questions and concerns are below.
>
> **Concern 1: Use of proxies**
>
> We agree with the reviewer: proxy-sensitive attributes can be problematic as they introduce another source of error into an already sensitive issue. Nonetheless, note that there are settings where (i) this is inevitable (when ground-truth attributes aren't available/recorded), or (ii) when subjects wish to withhold their ground-truth sensitive data (e.g. for privacy reasons). In both these cases, we show that one can nonetheless provide precise guarantees on maximum fairness violations; thus, these proxies can be helpful. We will make sure to discuss these concerns in depth, the pros and cons of using proxies, in our revised version.
>
> **Concern 2: Figures**
>
> Regarding the comment that “the proxies exhibit some error, albeit small.”, we note that we are referring to the misclassification error of proxies, which are precisely reported in Tables 1-3 in the main text, along with Tables 4-6 in the Appendix. We could turn these tables into figures, if necessary, but we thought the tabular format to be sufficient.
>
> We apologize for the lack of clarity in Figures 1-4. The x-axis refers to the group memberships, and the y-axis is the expected calibration error of each of the groups, $ECE(f,g)$. To make the figures more readable, we refer to the groups as $g_1, \dots, g_k$, but we realize that it may be unclear which group each $g_i$ is referring to. We will certainly clarify in this revised version!
>
> Additionally, the 2 lines refer to the actual violation (red) and upper bound (blue). We understand that the legend may be confusing. In the revised version, we will place “True Violation” directly above the dotted red line and “Upper Bound” directly above the blue line, making it clearer.
>
> **Additional comments**
>
> What is $v$ in the ECE definition?
>
> The value $v$ refers to every value in $[0,1]$ that the predictor can take. For example, let $v = 0.3$. Then, the inner term in the ECE is just $|E[g(X,Z)(0.3 - Y)|f(X)=0.3]|$. That is, the group-wise error for all points $X$ where $f(X)=0.3$. The expected calibration error takes the average of this quantity over all $v$, where $v$ is sampled from the distribution of predictions made by $f$.
>
> To be more clear, we will edit the manuscript with the following:
>
> "Central to evaluating MC is the expected calibration error (ECE) for a group $g \in \mathcal{G}$
>
> $$ECE(f,g) = E_{v \sim \mathcal{D}_f}[|E[g(X,Z)(f(X)−Y)|f(X) = v]|]$$
>
> where the outer expectation is over $v \sim \mathcal{D}_f$, the distribution of predictions made by the model $f$ under $\mathcal{D}$"
>
> We also thank you for catching the other typos! We will make sure to fix them in the revised manuscript.
>
> Finally, thank you for your questions and comments! We hope we have addressed them in a clear and satisfactory manner.

---

> > ### Comment · Reviewer_HMNE · 2025-04-02
> >
> > I would like to thank the authors for answering my concerns.

---

### Official Review · Reviewer_LB2V · 2025-03-17

**Overall Recommendation:** 2

**Summary:**

This paper explores the problem of evaluating and achieving multiaccuracy (MA) and multicalibration (MC) when sensitive group information is missing. The authors address this challenge by learning proxy functions to predict group membership without direct access to sensitive attributes.
1. Theoretical results: The paper establishes an upper bound on the evaluation error of MA and MC for the original groups in terms of the corresponding error on the proxy groups. Additionally, the authors propose an algorithm that post-processes a predictor to ensure multiaccuracy and multicalibration with respect to the proxy groups, which in turn guarantees these properties for the original groups.
2. Empirical results: Experimental results demonstrate that post-processing based on proxy groups effectively reduces worst-case MA and MC violations for the original groups.

**Claims And Evidence:**

1. Line 234 "Conversely, if the worst-case violations are large, this suggests that f may potentially be significantly biased or uncalibrated for certain groups g". Although the authors emphasizes the word "potentially", I still don't think it is a right claim to make in the theoretical section as they do not have a theorem that lower bound error on the original groups by the error of the proxy groups.
2. For the claim made by authors after theorem 5.1, they should also emphasize that they guarantee they get is for the so-called "worst-case" violation in their bound analysis, but does not necessarily mean that the violation gets smaller.
In general, the authors should be careful to make it clear that the worst-case violations does not mean the actually violation, and also this worst-case violations might not be tight upper bound for certain cases.

**Essential References Not Discussed:**

No.

**Experimental Designs Or Analyses:**

I have checked the soundness of the experiments, and the results appear solid. However, I have a few clarifications and suggestions:
1. In line 436, the statement "Notably, both models are approximately 0.03-multicalibrated with respect to the proxies" might be incorrect. Should this refer to the original groups instead of the proxies?
2. It would be helpful if the authors discussed how the predictor could be further calibrated with respect to the original groups in cases where it already exhibits small violations for the proxies. This could provide deeper insights into the limitations and effectiveness of the approach.
3. Additionally, I would like clarification on how the upper bound in the figure is determined. Does it take finite-sample analysis into account, or is it simply computed using the evaluation error? Providing details on this would improve the interpretability of the results.

**Methods And Evaluation Criteria:**

Learning proxy functions is a natural and intuitive approach for this problem, and the chosen benchmark/dataset is appropriate. However, I am uncertain about how the upper bound bar in the figure is determined, whether it takes finite sample analysis into consideration.

**Other Comments Or Suggestions:**

No.

**Other Strengths And Weaknesses:**

Strengths:
1. The writing is clear and well-structured, making the paper easy to follow.
2. This is the first work to study the use of proxy functions for achieving multiaccuracy (MA) and multicalibration (MC) in the setting where sensitive groups information is missing.
Weaknesses:
1. The paper lacks algorithmic innovation. Rather than introducing a new algorithm, the authors primarily adapt existing methods by applying them to proxy groups instead of the original groups. While this is a meaningful extension, it limits the novelty of their technical contributions.

**Questions For Authors:**

1. I would like clarification on how the upper bound in the figure is determined. Does it take finite-sample analysis into account, or is it simply computed using the evaluation error?

**Relation To Broader Scientific Literature:**

The algorithm used to achieve multiaccuracy (MA) and multicalibration (MC) in this paper is based on the work of Roth (2022) and Globus-Harris et al. (2023), meaning the authors do not introduce a new algorithm. Their primary contribution lies in addressing the scenario where the sensitive attributes that define the groups are missing. While fairness under incomplete sensitive data has been studied in the context of other fairness metrics, this paper is the first to explore the problem specifically for MA and MC. Their approach leverages proxy functions to address this challenge.

**Theoretical Claims:**

I have checked all the proofs, and they appear to be correct. However, some theorems lack clear interpretations. As I previously pointed out, their theorems suggest that the "worst-case" error decreases, but the worst-case analysis may not be tight. Additionally, reducing worst-case violations does not necessarily imply a reduction in the true violation.

For instance, the bound in Theorem 5.3 might not be tight. The authors seem to suggest that achieving an
epsilon error would require 1/epsilon^8 iterations, which appears overly pessimistic. I recommend that the authors carefully verify the result in Globus-Harris et al. (2023).

Moreover, it would be beneficial if the authors included a finite-sample analysis.

---

> ### Author Rebuttal · Authors · 2025-03-31
>
> Thank you for the thoughtful review! Our responses to your questions and concerns are below.
>
> **Concern 1: General Concerns around Worst-Case Violations**
>
> As you correctly state, we establish upper bounds on the MA/MC violations across the true groups in terms of the violations on the proxies, the proxy errors, and the $MSE$ of $f$. We would like to add that
>
> 1) We have established lower bounds using identical techniques for MA (and we believe the same hold for MC); we will include them in the revised version.
> 2) The bounds are **tight**, i.e. there exists a distribution over $(f, Y, g, \hat{g})$ such that the bounds hold with **equality**:
>
> W.l.o.g., consider the AE violation for a group $g$. Consider first $MSE(f) \leq P(\hat{g} \neq g)$, so the upper bound is
>
> $$
> |AE(f, g)| \leq |AE(f, \hat{g})| + \sqrt{MSE(f)}\cdot\sqrt{P(\hat{g} \neq g)}.
> $$
>
> Assume also that $MSE(f) > 0$ and $P(\hat{g} \neq g) > 0$ (i.e. not perfect predictors). The bound holds with equality if the data-generating process satisfies:
>
> 1) $P(\hat{g} \neq g) = P(\hat{g} = 0, g=1)$
> 2) $f-Y  = \lambda\cdot(g - \hat{g})$ where $\lambda = \sqrt{\frac{MSE(f)}{P(\hat{g} \neq g)}}$
>
> On the other hand, when $MSE(f) > P(\hat{g} \neq g)$, the bound is
>
> $$
> |AE(f, g)| \leq |AE(f, \hat{g})| + P(\hat{g} \neq g),
> $$
>
> With analogous (but slightly more involved) steps, one can construct a distribution where this upper bound holds with equality.
>
> We will include all these points in the revised version. Given these two facts, we believe that our claim —“If the worst-case violations are large, this suggests that $f$ may potentially be significantly biased or uncalibrated for certain groups”—is reasonable and hope the reviewer agrees.
>
> Additionally, we agree that it is important to emphasize that Theorem 5.1 applies to the worst-case violation, and not the true violation. In our setting, the true violation cannot be evaluated; thus, one cannot provably reduce it. Yet, we can minimize the upper bound. The reviewer is correct that this will not reduce the actual violation, but we argue this is still an effective way to provide meaningful guarantees — as demonstrated in the experiments. We can modify $f$ such that the upper bound is less than $\alpha$,  concluding that the *true violation* is also less than $\alpha$. We will make sure to clarify this distinction in the revised manuscript.
>
> **Concern 2: Figures/Finite Sample Analysis**
>
> Recall the upper bound is a function of $MSE(f)$, $err(\hat{g})$, and $ECE(f, \hat{g})$. Here $f$ is the output of a learning algorithm that takes a training sample $S_{train}$, and the modified $f$ is the output of running the algorithm on a calibration set $S_{cal}$. To compute the upper bound, we simply compute sample estimates of all 3 quantities on a fixed held-out set and report the average over five train/calibration splits. Note that stating the theoretical results in terms of the distributional quantities is standard practice in this area of work [1,2,3].
>
> We do not include a finite sample analysis in the manuscript but refer the readers to [2] where a complete finite sample analysis is done. The analysis repeatedly uses Chernoff bounds to show that, as long as the algorithms run on a finite sample of $n$ i.i.d samples from $\mathcal{D}$, then the guarantees carry over to the true distribution with high probability when $n$ is sufficiently large. We will happily include the main results from [2] and explain their application to our setting in our revised appendix.
>
> **Concern 3: Lack of Algorithmic Innovation**
>
> While we do not introduce a new algorithm, we do not think this is a weakness. When trying to extend various fairness guarantees via proxies, other works have needed to provide new algorithms to control either the true violation or upper bounds [4,5] because the theory demanded it. Our theory clearly shows that a new algorithm is **not** needed, which we believe to be an elegant contribution.
>
> **Additional comments**
>
> 1) Thank you for pointing out the typo in Theorem 5.3! It should be $T < \frac{4}{\alpha^2}$ rounds. We will correct this.
> 2) Line 436 is correct. Since the models are already highly multicalibrated across the proxies, correcting more provides a minimal reduction in our bounds.
>
> Thank you for your questions and comments! We hope we have addressed them in a clear and satisfactory manner.
>
> [1] "Multicalibration for Confidence Scoring in LLMs", Detomasso et al. ICML 2024.
> [2] "Uncertain: Modern Topics in Uncertainty Estimation", Roth 2022.
> [3] "Multicalibration as Boosting for Regression", Globus-Harris et al. ICML 2023
> [4] "Estimating and Controlling for EOD via Sensitive Attribute Predictors", Bharti et al. NeurIPS 2023.
> [5] "Multiaccurate Proxies for Downstream Fairness", Diana et al. FaccT 2022.

---

> > ### Comment · Reviewer_LB2V · 2025-04-08
> >
> > I would like to thank the reviewers for addressing most of my questions. Overall, I feel that my concerns were generally well answered. I will maintain my current score for now, as I mentioned before the algorithmic novelty and theoretical contributions to be relatively straightforward, but I appreciate the authors' efforts in improving the paper and will discuss with AC and other reviewers in the discuss period.

---

### Decision · Program_Chairs · 2025-05-01

**Decision:**

Accept (poster)

**Comment:**

The paper considers the problem of ensuring multiaccuracy and multicalibration using "proxy" groups, i.e., when access to exact group-denoting labels are not available, but a function that "predicts" group measurement is available. Overall, the results are a relatively direct extension of existing work in multiaccuracy and multicalibration, with the additional 'twist' of lack of access to group-denoting information.

The reviewers were split on the merits of the paper. Reviewer LB2V found the novelty somewhat limited, with the proposed algorithms being straightforward extensions of, e.g., boosting algorithms existing in the literature. Reviewer hLPe also noted that the results could be improved to have greater relevance for practitioners (e.g., when to improve the model vs. the proxy scorer).

The other two reviewers were more positive, though reviewer VC5A noted that the paper misses comparison with basic data imputation techniques.

Overall, I found the messaging of this paper confusing. One of the main points frequently made in the multiaccuracy and multicalibration literature is that pre-defined population groups are not required -- even if only defined via proxy variables. The goal in this line of work is to automatically outline "computationally identifiable groups" where the model errs, instead of relying on pre-defined group attributes (variable Z in the paper). I understand that the authors still optimize over a group of functions over X and Z, though they also assume a 'proxy developer' with explicit access to (X,Z) samples. This seems to go against the spirit of the multiaccuracy and multicalibration literature.

Though the paper is a relatively direct extension of existing work on multiaccuracy and multicalibration, it still presents a solid theoretical development and experiments, as recognized by the reviewers. However, the limited novelty limits the potential impact of the contribution. I encourage the authors to take the reviewers comments into account.